# AutoMS: Multi-Agent Evolutionary Search for Cross-Physics Inverse Microstructure Design

Zhenyuan Zhao [* 1]   Yu Xing [* 1]   Tianyang Xue [2]   Lingxin Cao [1]   Xin Yan [1]   Lin Lu [1]

## Abstract

Designing microstructures with coupled cross-physics objectives is a fundamental challenge where traditional topology optimization is often computationally prohibitive and deep generative models frequently suffer from physical hallucinations. We introduce AutoMS, a multi-agent neurosymbolic framework that reformulates inverse design as an LLM-driven evolutionary search. AutoMS leverages LLMs as semantic navigators to decompose complex requirements and coordinate agent workflows, while a novel Simulation-Aware Evolutionary Search (SAES) mechanism handles low-level numerical optimization via local gradient approximation and directed parameter updates. This architecture achieves a state-of-the-art 83.8% success rate on 17 diverse cross-physics tasks, significantly outperforming both traditional evolutionary algorithms and existing agentic baselines. By decoupling open-ended semantic orchestration from simulation-grounded numerical search, AutoMS provides a robust pathway for navigating complex physical landscapes that remain intractable for standard generative or purely linguistic approaches.

## 1 Introduction

The automated discovery of microstructures, complex geometric architectures that determine the macroscopic properties of materials, remains a frontier challenge in AI for Science (Bertoldi et al., 2017; Vyatskikh et al., 2018). From lightweight aerospace lattices to biocompatible implants, the ability to inverse-design structures with specific, often conflicting, cross-physics properties promises transformative engineering breakthroughs (Surjadi & Portela, 2025; Ha et al.,

---
[*]Equal contribution  [1]Shandong University, Qingdao, China [2]The University of Hong Kong, Hong Kong, China. Correspondence to: Lin Lu <llu@sdu.edu.cn>.

*Proceedings of the 43rd International Conference on Machine Learning*, Seoul, South Korea. PMLR 306, 2026. Copyright 2026 by the author(s).

2023; Schumacher et al., 2015; Zheng et al., 2014). However, this task represents a non-trivial optimization problem: the search space is high-dimensional and discontinuous, while the objective function—derived from coupled cross-physics simulations—is non-differentiable and expensive to evaluate (Takezawa et al., 2014).

Existing computational approaches generally fall into two paradigms, each with distinct limitations: Topology Optimization (TO) offers principled derivation but struggles to escape local optima in non-differentiable landscapes (Meng et al., 2026; Takezawa et al., 2014), while Deep Generative Models (DGMs) operate as "one-shot" probabilistic mappings that, despite their exploratory power, lack the inherent reasoning capability to negotiate conflicting "cross-physics" constraints (Vlassis & Sun, 2023; Regenwetter et al., 2022; Xue et al., 2025). Consequently, DGMs often suffer from physical hallucination, generating visually plausible but physically invalid structures (Behzadi et al., 2024; Bastek & Kochmann, 2023), because they optimize for statistical likelihood rather than physical validity (Mazé & Ahmed, 2023). To bridge this "validity gap," we contend that a paradigm shift is necessary: moving from "one-shot" generation to a closed-loop iterative refinement process. Evolutionary search provides a natural solution to this challenge by leveraging simulation feedback to progressively eliminate invalid designs (Deb et al., 2002; Mouret & Clune, 2015). However, traditional evolutionary operators lack semantic understanding. Compounding this challenge is an interaction bottleneck: the fuzzy semantics in human natural language leads to a misalignment between the designer's "want" and the actual "get". Conversely, while LLMs empowered by reasoning frameworks (Yao et al., 2023; Gao et al., 2023) provide this capacity, naive single-agent approaches fail due to their open-loop nature. The critical solution, therefore, is to ground LLM semantic creativity within a rigorous, closed-loop physical validation process.

To address these limitations, we introduce AutoMS, a multi-agent neuro-symbolic framework that reformulates inverse design as an LLM-driven evolutionary search, as illustrated in Figure 1. Leveraging MIND (Xue et al., 2025), originally designed for single-physics inverse design, as our foundational generator, AutoMS establishes a systematic

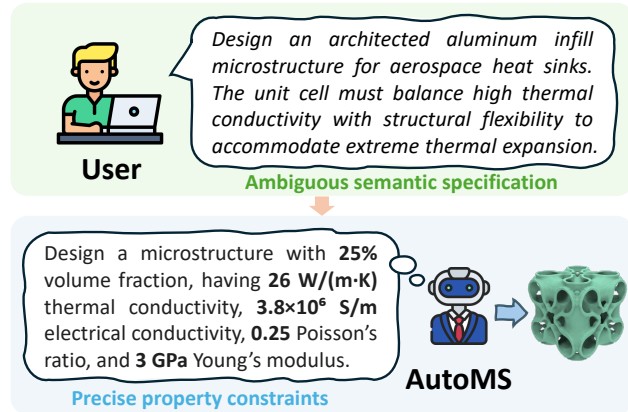

*Figure 1.* **The AutoMS workflow: From abstract intent to concrete microstructure design.** The system takes ambiguous semantic specifications as input and operationalizes them into quantifiable targets. Through a simulation-aware evolutionary loop, AutoMS iteratively refines the design candidates, resulting in optimized, topologically coherent microstructures that satisfy the initial cross-physics demands.

pathway to bridge single-physics generation with rigorous cross-physics requirements.

AutoMS adopts a two-layer architecture, with an Orchestration Layer that translates natural-language intent into quantifiable targets and coordinates agents, and an Optimization Layer that runs a simulation-grounded evolutionary loop. In the Optimization Layer, the Generator uses MIND to propose candidates and the Simulator evaluates them under cross-physics constraints. Simulation-Aware Evolutionary Search (SAES) serves as the core controller of this layer. It uses cross-physics feedback to guide mutation or crossover exploration, filtering physically invalid geometries and steering the search toward feasible designs, while coordinating Pareto-driven selection and simulation-informed updates in the parameter space to meet target specifications under conflicting objectives.

Our contributions are as follows:

- **Neuro-Symbolic Framework**: We propose AutoMS, the first framework that operationalizes LLM agents as semantic evolutionary operators to automate inverse design under coupled cross-physics constraints.
- **Simulation-Aware Evolutionary Search (SAES)**: We introduce SAES, a closed-loop optimization strategy that leverages simulation feedback to guide agent decision-making, significantly reducing physical hallucinations compared to standard generative methods.
- **Empirical State-of-the-Art**: Across a benchmark of 17 diverse tasks, AutoMS achieves a 83.8% success rate, nearly doubling the performance of traditional NSGA-II (43.7%) and outperforming strong numerical baselines such as BayesOpt (65.0%) and CMA-ES (66.7%). Fur-

thermore, the hierarchical orchestration reduces wall-clock time by 23.3% compared to ReAct-based LLM baselines by pruning physically invalid search paths.

## 2 Related Work

### 2.1 Physics-Constrained Microstructure Generation

Early computational microstructure design focused on parameterized periodic architectures, which restrict the design space (Ashby, 2006; Zheng et al., 2014). Topology optimization expanded this space via gradient-based discovery of complex geometries through SIMP and level-set methods (Bendsøe & Sigmund, 2004; Yulin & Xiaoming, 2004), with multiscale extensions enabling concurrent optimization of macroscopic structures and underlying microstructures (Xia & Breitkopf, 2014; Wu et al., 2021). Despite these advances, such methods typically target single-physics objectives, require substantial expert intervention, and rely on repeated high-fidelity simulations, limiting scalability and interactive exploration.

Inverse homogenization reframes microstructure design as an inverse problem, in which unit-cell geometries are optimized to match prescribed effective properties (Sigmund, 1994). Although theoretically grounded, these methods often suffer from ill-posedness, non-convex optimization landscapes, and limited integration of manufacturing or cross-physics constraints. Data-driven extensions accelerate property evaluation by querying databases of pre-computed microstructures (Bessa et al., 2017; Sardeshmukh et al., 2024), but their reliance on existing samples restricts extrapolation and complicates multi-objective design involving competing performance requirements.

Recent advances in artificial intelligence reshape the paradigms for inverse microstructure design (Zheng et al., 2025; Van et al., 2025). Generative models, including variational autoencoders and generative adversarial networks, enable learned representations of microstructure spaces and synthesis of novel geometries (Sardeshmukh et al., 2024; Fokina et al., 2020). Diffusion-based models further improve generation quality and controllability, and conditional diffusion has been applied to inverse design tasks targeting mechanical, thermal, and acoustic properties (Vlassis & Sun, 2023; Li et al., 2025b; Baishnab et al., 2025). The MIND framework (Xue et al., 2025) advances this direction by jointly encoding geometric and physical attributes in hybrid latent representations.

Complementary to generative approaches, neural operators provide efficient surrogate models for property evaluation (Li et al., 2020; Lu et al., 2021), reducing the computational cost of iterative optimization. However, most existing learning-based methods operate in open-loop settings, rely on surrogate approximations rather than direct

physical verification, and struggle to robustly enforce coupled cross-physics constraints. These limitations motivate closed-loop, physics-constrained frameworks capable of autonomously exploring inverse microstructure design spaces under conflicting physical objectives.

## 2.2 Multi-Agent LLMs for Physics-Grounded Design

Large language models (LLMs) have recently shown promising capabilities in scientific workflows, particularly in knowledge reasoning and tool orchestration (Wang et al., 2023). Domain-specific systems such as ChatMOF (Kang & Kim, 2024) and ChemHAS (Li et al., 2025c) demonstrate how language interfaces can coordinate complex computational tools. However, most existing systems focus on property prediction, data retrieval, or isolated workflow automation, rather than end-to-end design generation with physics-based validation. MetaGen (Makatura et al., 2025) leverages the strong coding and reasoning capabilities of LLMs for metamaterial generation. However, LLMs lack intrinsic awareness of physical laws (Ali-Dib & Menou, 2024). Consequently, numerical simulation remains essential for ensuring physical consistency, motivating the integration of LLMs with simulation-driven agent systems.

The shift from monolithic models to collaborative multi-agent architectures has expanded the applicability of LLMs in scientific computing (Zheng et al., 2025; Wang et al., 2023). General-purpose frameworks such as AutoGen (Wu et al., 2024), AutoAgents (Chen et al., 2024a), and MetaAgent (Zhang et al., 2025b) enable coordinated problem decomposition through structured agent interactions (Barbosa et al., 2024). These paradigms have been applied to autonomous chemical synthesis (Boiko et al., 2023), metal–organic discovery (Kang & Kim, 2024), and physics-informed neural network automation (Wuwu et al., 2025).

Recent efforts further extend multi-agent LLMs to physics-based simulations. Lang-PINN (He et al., 2025) maps natural language specifications to PINN architectures, while MooseAgent (Zhang et al., 2025a) and MetaOpenFOAM (Chen et al., 2024b) automate finite element and CFD workflows. In materials science, hybrid GNN–LLM systems have been proposed for rapid alloy design (Ghafarollahi & Buehler, 2025). Related studies also explore simulation parameterization (Xia et al., 2024), end-to-end atomistic simulations (Vriza et al., 2026), and cooperative optimization in cyber-physical systems (Chen & He, 2025).

Despite these advances, multi-agent LLM systems have not been systematically applied to inverse geometry or microstructure design from ambiguous semantic intent. Existing approaches typically operate in open-loop or single-physics settings, lacking validation to enforce physical constraints and struggling with multi-objective optimization under conflicting requirements (Li et al., 2025a). In contrast, our work performs end-to-end inverse design from semantic descriptions to geometry candidates validated through closed-loop cross-physics simulation. Our approach enables physically grounded, multi-objective optimization rather than purely generative or tool-centric systems.

## 3 Methodology

### 3.1 Problem Formulation

We formulate the cross-physics microstructure design as an optimization problem over a mechanical conditioning space $\mathcal{X}$. Specifically, we treat the mechanical properties (e.g., Elastic Modulus $E$, Poisson's ratio $\nu$) as the design variables $\mathbf{x} \in \mathcal{X}$, where $\mathbf{x}$ is a continuous conditioning vector in latent mechanical space, which serve as the input conditions for the generative process.

Let $\mathcal{G}_{\text{MIND}}$ denote the pre-trained MIND (Xue et al., 2025) generative model, which maps a set of mechanical conditions $\mathbf{x}$ to a microstructure geometry $\mathbf{m} \in \Omega$, and $\Omega$ denotes the admissible microstructure geometry space. Subsequently, a high-fidelity simulator $\mathcal{S}$ evaluates the generated structure $\mathbf{m}$ to obtain its cross-physics performance (e.g., thermal conductivity $\kappa$), which serves as the objective.

Our objective is to identify the optimal mechanical conditioning parameters $\mathbf{x}^*$ that induce a microstructure maximizing the expected cross-physics performance:

$$\mathbf{x}^* = \arg\max_{\mathbf{x} \in \mathcal{X}} \mathbb{E}_{\mathbf{m} \sim \mathcal{G}_{\text{MIND}}(\mathbf{x})}[\mathcal{S}(\mathbf{m})]. \tag{1}$$

This formulation shifts the design paradigm from directly manipulating geometric voxels to optimizing the latent mechanical conditions, effectively navigating the complex inverse mapping between mechanical inputs and cross-physics outputs.

### 3.2 The AutoMS Framework

To navigate the non-differentiable and stochastic optimization landscape defined in Equation (1), we introduce **AutoMS**, a neuro-symbolic framework that orchestrates specialized agents via a **Two-Stage Hierarchical Architecture** (Figure 2). AutoMS decouples high-level semantic planning from low-level numerical search. This structure grounds the semantic flexibility of LLMs within the rigorous constraints of physical laws through two specialized layers:

**Orchestration Layer.** Governed by the **Manager**, this layer manages the "Semantic-to-Parametric" translation and high-level task supervision. Upon receiving a natural language request, the Manager invokes the **Parser**. Leveraging the open-source GraphRAG framework (Edge et al., 2024), the Parser resolves vague qualitative descriptions into quantifiable targets within the mechanical conditioning space $\mathcal{X}$.

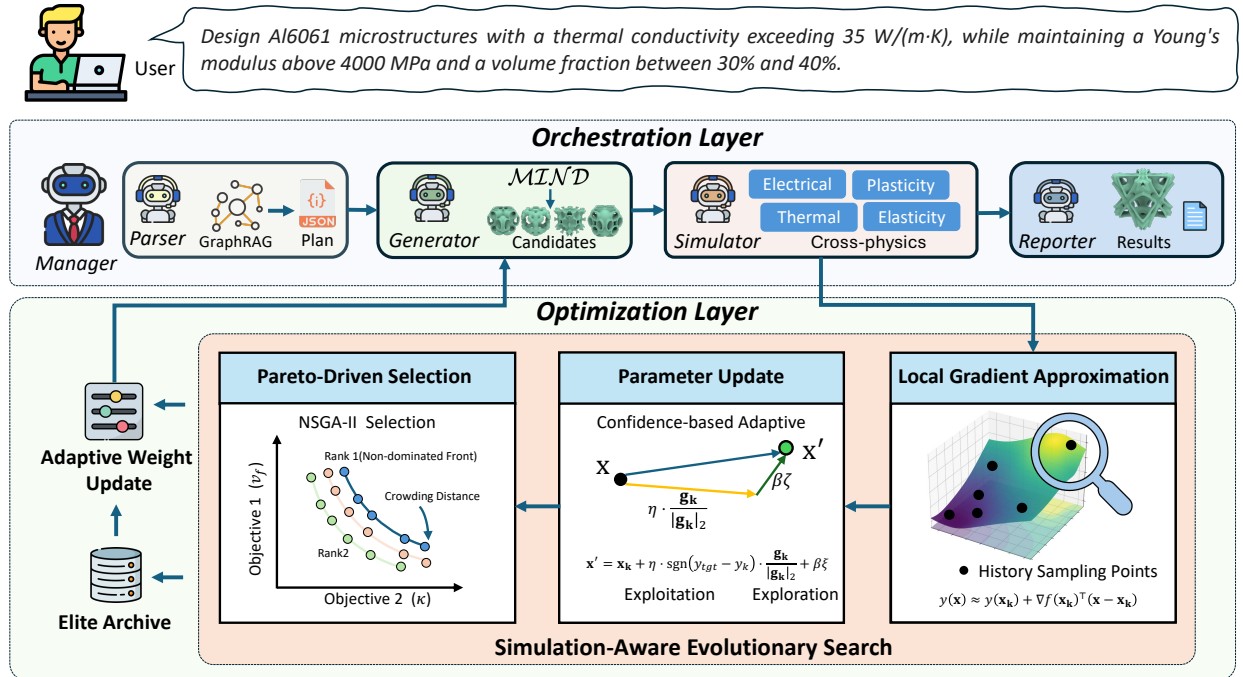

*Figure 2.* **The AutoMS Multi-Agent Architecture for Cross-Physics Inverse Design.** The framework comprises two hierarchical layers: Orchestration Layer: Governed by a Manager Agent, this layer coordinates the Parser (leveraging GraphRAG) to operationalize semantic intent into parametric targets, while the Generator, Simulator, and Reporter agents handle geometry synthesis, physical validation, and results synthesis, respectively. Optimization Layer: Functioning as the numerical engine, it executes Simulation-Aware Evolutionary Search (SAES) through four mathematical phases: Local Gradient Approximation (Perception) via Weighted Least Squares, Parameter Update (Action) using gradient-guided navigation, and Pareto-Driven Selection combined with Adaptive Weight Updates (Integration) to resolve conflicting objectives and break local optima.

The **Generator** then synthesizes geometries via MIND, the **Simulator** performs FEA validation (details in Appendix D), and the **Reporter** synthesizes the final recommendations. The detailed system prompts defining the specific behaviors and constraints for all agents are provided in Appendix A.1. By flagging high-level exceptions, including violations of physical feasibility constraints, the Manager counteracts the hallucinations inherent to open-loop LLM reasoning.

**Optimization Layer: Simulation-Aware Evolutionary Search (SAES).** Functioning as the computational engine, this layer executes the SAES algorithm to iteratively refine the conditioning parameters $\mathbf{x}$. This layer operates through four specialized numerical phases: **Local Gradient Approximation (Perception)** estimates the search direction from historical simulation data using Weighted Least Squares; **Parameter Update (Action)** applies gradient-guided navigation with adaptive step sizes to drive $\mathbf{x}$ toward target properties; **Pareto-Driven Selection (Integration)** maintains population diversity and preserves elite candidates through non-dominated sorting, and Adaptive Weight Update dynamically rebalances objective priorities to break through local optima. This numerical backbone effectively bridges the gap between high-level semantic intent and low-level physical validity.

## 3.3 Simulation-Aware Evolutionary Search (SAES)

The core of the Optimization Layer is SAES, designed to navigate the non-differentiable landscape of cross-physics simulations. SAES constructs a local gradient approximation from simulation feedback to guide MIND's conditioning parameters $\mathbf{x}$. The process consists of three phases: Perception (Local Gradient Approximation), Action (Gradient-Guided Parameters Update), and Integration (Pareto-Driven Selection & Adaptive Update).

**Local Gradient Approximation (Perception)** Since the cross-physics simulator $\mathcal{S}$ is a black-box oracle without analytical gradients, we cannot directly compute $\nabla f(\mathbf{x})$. To overcome this, we construct a **simulation-aware local gradient** by estimating local gradients from historical data. Let $\mathcal{D} = \{(\mathbf{x}_i, y_i)\}_{i=1}^N$ denote the history of evaluated designs, where $\mathbf{x} \in \mathcal{X}$ is the mechanical conditioning vector and $y_i = y(\mathbf{x}_i)$ is the simulated scalar response. We use $y(\mathbf{x})$ as the response notation throughout, and define the local objective as $f(\mathbf{x}) := y(\mathbf{x})$. To capture the local geometry around the current candidate $\mathbf{x}_k$, we employ a sliding window strategy, selecting $M$ nearest neighbors $\mathcal{S}_k \subset \mathcal{D}$.

We approximate the local gradient via a first-order Taylor expansion centered at $\mathbf{x}_k$:

$$y(\mathbf{x}) \approx y(\mathbf{x}_k) + \nabla f(\mathbf{x}_k)^\top (\mathbf{x} - \mathbf{x}_k), \qquad (2)$$

where $\mathbf{g}_k = \nabla f(\mathbf{x}_k)$ is the estimated local gradient. We solve for $\mathbf{g}_k$ by minimizing the Weighted Least Squares (WLS) error:

$$\mathbf{g}_k = \arg\min_{\mathbf{g}} \sum_{(\mathbf{x}_j, y_j) \in \mathcal{S}_k} w_j \cdot \left( y_j - y_k - \mathbf{g}^\top (\mathbf{x}_j - \mathbf{x}_k) \right)^2. \qquad (3)$$

To ensure robustness against the non-stationary nature of the search process, the weight $w_j$ is designed as a composite metric of both spatial proximity and temporal freshness:

$$w_j = \underbrace{\frac{1}{1 + \|\mathbf{x}_j - \mathbf{x}_k\|^2}}_{w_{\text{dist}}(\mathbf{x}_j, \mathbf{x}_k)} \cdot \underbrace{\exp\left( -\lambda_t \frac{N - t_j}{N} \right)}_{w_{\text{time}}(t_j)}. \qquad (4)$$

Here, $w_{\text{dist}}$ assigns higher confidence to neighbors geometrically closer to the query point $\mathbf{x}_k$, ensuring geometric locality. The term $w_{\text{time}}$ introduces a temporal decay based on the iteration index $t_j$ of the sample $\mathbf{x}_j$. By incorporating the decay rate $\lambda_t$, this mechanism effectively down-weights stale data points (where $N - t_j$ is large) that may exert a detrimental influence due to distribution shifts, while prioritizing recent observations that provide a more accurate reflection of the current optimization landscape.

**Gradient-Guided Parameter Update (Action)** To accelerate convergence, we introduce a Gradient-Guided Parameter Update that transforms the random updating into a directed navigation:

$$\mathbf{x}' = \mathbf{x}_k + \eta \cdot \text{sgn}(y_{tgt} - y_k) \cdot \frac{\mathbf{g}_k}{\|\mathbf{g}_k\|_2} + \beta \xi, \qquad (5)$$

where $\eta$ is the adaptive step size and $\xi$ represents Gaussian exploration noise. $\beta$ denotes the exploration noise coefficient. The term $\text{sgn}(y_{tgt} - y_k) \in \{-1, 1\}$ encodes physics-aware intuition: it acts as a directional switch that drives the search along the gradient to boost properties when undershooting, or against the gradient to reduce them when overshooting. This mechanism explicitly corrects the mechanical inputs $\mathbf{x}$ toward the cross-physics target based on simulation trends.

**Pareto-Driven Selection & Adaptive Update (Integration)** The final phase, Integration, serves two critical functions: preserving high-quality candidates and rectifying search stagnation.

**Pareto-Driven Selection**: We employ the non-dominated sorting mechanism of NSGA-II to stratify the population

into Pareto frontiers $\mathcal{F}_1, \mathcal{F}_2, \ldots$. To prevent mode collapse, we apply crowding distance sorting to maintain diversity in the conditioning space. An Elite Archive $\mathcal{A}$ is maintained to preserve valid designs that satisfy physical constraints, ensuring monotonicity in performance improvement.

**Adaptive Weight Update**: To address cases where the parameter $\mathbf{x}$ stagnates due to conflicting objectives (e.g., maximizing thermal conductivity while minimizing volume fraction), we implement an adaptive guidance strategy. We monitor the improvement ratio $\gamma_j$ of each objective over a sliding window. If stagnation is detected ($\gamma_j \approx 0$), the scalarization weights $w$ governing the optimization utility are dynamically rebalanced:

$$w_j^{(t+1)} = \text{clip}\left( w_j^{(t)} \cdot (1 + \delta(\gamma_j)), 0.1, 2.0 \right), \qquad (6)$$

where $\delta$ is a penalty factor. This weight adjustment effectively rotates the aggregated gradient direction for the subsequent Perception phase, compelling the search to break out of local optima and explore under-optimized regions of the Pareto frontier.

To illustrate the practical coordination of these components, we provide a concrete workflow example in Appendix A.2. Additionally, detailed implementation specifications and SAES hyperparameters are documented in Appendix B.

## 4 Experimental Setup

**Hardware and Software Configuration.** All experiments were conducted on a high-performance computing cluster equipped with two NVIDIA A40 GPUs (48 GB VRAM each), 512 GB system RAM, and a 64-core AMD EPYC 7763 processor. The AutoMS framework is implemented in Python 3.10, with simulation engines powered by JAX-FEM (Xue et al., 2023) for cross-physics analysis. LLM inference utilizes DeepSeek-V3.2 as the default backbone, with additional experiments on GPT-4o, GPT-5.2, Claude Haiku 4.5, and Qwen3-Max for robustness evaluation.

**Benchmark Tasks.** We designed a comprehensive suite of 17 tasks covering four orthogonal dimensions to evaluate the framework's versatility:

- **Material Diversity**: Encompassing high-performance metals (Al6061, Ti6Al4V, AlSi10Mg) and engineering polymers (ABS, TPU).
- **Optimization Modality**: Including single-objective (5 tasks), multi-objective with 2–5 competing constraints (8 tasks), and Pareto frontier exploration (4 tasks).
- **Physical Coupling**: Ranging from isolated physical fields to full tri-field coupling, supplemented by plasticity simulations for nonlinear deformation analysis.
- **Difficulty Stratification**: Tasks are classified into Easy

(3), Medium (8), and Hard (6) based on objective cardinality, proximity to physical limits, and simulation fidelity.

**Evaluation Metrics.** To systematically quantify the performance of AutoMS, we assess the framework across three critical dimensions:

- **Task Completion**: Measures the system's reliability in meeting strict design requirements, tracked via *Success Rate (SR)* for fully valid solutions and *Constraint Satisfaction Rate (CSR)* for partial objective fulfillment.
- **Solution Quality**: Evaluates the physical precision and multi-objective trade-offs of the generated microstructures. Metrics include *Mean Relative Error (MRE)* for accuracy, *Best Property Match (BPM)* for Pareto alignment, and a composite *Quality Score (QS)* for holistic ranking.
- **Efficiency**: Quantifies the computational cost required for convergence, measured by *Iterations (Iter)* and total execution *Time* (s) excluding plasticity simulations.

Detailed mathematical definitions and calculation formulas for all metrics are provided in Appendix E. All experimental results are averaged over four independent runs to ensure statistical robustness.

**Baselines.** To benchmark AutoMS against established paradigms and isolate the contributions of its hierarchical neuro-symbolic architecture, we compare it with representative baselines covering traditional numerical optimization and monolithic LLM reasoning:

- **NSGA-II** (Deb et al., 2002): A classic multi-objective genetic algorithm. It optimizes the *conditioning parameters* $(E, G, \nu)$ passed to the MIND through evolutionary operators, using the same simulation engines as AutoMS but without natural language reasoning or agent collaboration.
- **ReAct** (Yao et al., 2023): A single-agent Large Language Model baseline based on the ReAct paradigm. It performs iterative reasoning and action planning to solve the design task. Unlike AutoMS, it operates as a solitary entity without role specialization or collaborative dialogue.
- **Bayesian Optimization** (BayesOpt) (Snoek et al., 2012): A standard black-box optimizer operating on the same conditioning space and simulation feedback.
- **CMA-ES** (Hansen & Ostermeier, 2001): A strong covariance-adaptation evolutionary strategy baseline under the same simulator budget.
- **Single-Agent + SAES** (S-SAES): A single-agent variant using the same SAES numerical update but without multi-agent role decomposition.
- **CoT + SAES** (C-SAES) (Wei et al., 2022): A chain-of-thought single-agent variant coupled with SAES updates.

For a fair comparison, all methods (AutoMS and baselines) invoke the identical microstructure generation model and

*Table 1.* **Quantitative Benchmark Results.** Comprehensive comparison of AutoMS against numerical and agentic baselines in a unified table. The framework is evaluated across three critical dimensions: Task Completion, Solution Quality and Efficiency.

| Method | Completion | | Solution Quality | | | Efficiency | |
|---|---|---|---|---|---|---|---|
| | SR ↑ | CSR ↑ | MRE ↓ | BPM ↑ | QS ↑ | Iter ↓ | Time (s) ↓ |
| NSGA-II | 43.7% | 53.3% | 0.4276 | 51.7% | 49.8 | 48.8 | 5550.2 |
| ReAct | 53.3% | 53.6% | 0.0345 | 67.2% | 61.2 | 10.0 | 2843.9 |
| BayesOpt | 65.0% | 55.6% | 0.0194 | 85.6% | 73.7 | 39.2 | 3982.9 |
| CMA-ES | 66.7% | 55.7% | 0.0241 | 84.7% | 73.7 | 40.8 | 4202.3 |
| S-SAES | 69.2% | 66.4% | 0.0172 | 86.5% | 78.2 | **9.6** | **1743.0** |
| C-SAES | 74.1% | **70.2%** | 0.0227 | 90.5% | 81.9 | 16.7 | 2980.8 |
| **AutoMS** | **83.8%** | 59.6% | **0.0140** | **94.2%** | **82.4** | 14.4 | 2180.5 |

cross-physics simulation engine. This standardization ensures that performance divergences are strictly attributable to the efficiency of the decision-making paradigm, ranging from evolutionary approaches and standalone LLM reasoning to collaborative neuro-symbolic search, rather than varying fidelities in synthesis or validation.

# 5 Experimental Results

## 5.1 Comparative Performance Analysis

**Superiority in Validity and Precision.** As summarized in Table 1, AutoMS achieves a state-of-the-art Success Rate (83.8%), significantly outperforming traditional evolutionary algorithms like NSGA-II (43.7%) and black-box optimizers such as CMA-ES (66.7%).

The performance disparity highlights the limitations of single-paradigm approaches. While purely numerical methods (NSGA-II, CMA-ES) frequently stall in the discontinuous search space due to a lack of informed semantic initialization, the monolithic ReAct baseline (53.3% SR) suffers from open-loop reasoning failures where generated designs lack physical grounding.Crucially, the comparison between AutoMS and its ablation variants, Single-Agent + SAES (69.2% SR) and CoT + SAES (74.1% SR), provides empirical evidence for the necessity of the multi-agent orchestration. The 14.6% SR gap between the full framework and the single-agent variant demonstrates that the success of AutoMS is not solely derived from the SAES algorithm. Instead, it stems from the specialized role decomposition.

Beyond categorical success, AutoMS exhibits exceptional solution precision. It achieves a Mean Relative Error (MRE) of 0.0140, an order of magnitude lower than NSGA-II (0.4276). With a Best Property Match (BPM) score of 94.2%, the framework proves its capability to not only locate feasible regions but also precisely converge onto the target physical specifications.

**Efficiency via Intelligent Orchestration.** While multi-agent systems often incur higher latency due to inter-agent

communication, AutoMS reduces the total wall-clock time by 23.3% compared to ReAct. This efficiency gain is attributable to the hierarchical two layer architecture. By filtering physically invalid requests and pruning the search space before engaging the expensive high-fidelity simulator, the Orchestration Layer acts as an effective computational gatekeeper, preventing the wasteful simulation cycles that plague the unguided ReAct baseline.

Detailed task-specific performance and visual trajectories are documented in Appendix C.

## 5.2 Cross-Physics Fidelity and Structural Analysis

To evaluate the handling of conflicting constraints, we analyze Task 1 (Two-Physics Precise) and Task 2 (Cross-Physics Precise), which require balancing inverse physical properties in Table 2.

*Table 2.* **Cross-Physics Fidelity Analysis on Representative Tasks.** Quantitative validation on Task 1 (Two-Physics Precise) and Task 2 (Cross-Physics Precise). Err is defined in Appendix E.2.

| Task | Property | Target | ReAct | | NSGA-II | | AutoMS | |
|---|---|---|---|---|---|---|---|---|
| | | | Val | Err% | Val | Err% | Val | Err% |
| Task 1 | $\kappa$ (W/(m · K)) | 0.035 | 0.0220 | -37.1 | 0.0554 | +58.3 | **0.0340** | **-2.9** |
| | $E$ (MPa) | 110 | 125.8 | +14.4 | 484.7 | +340.6 | **107.4** | **-2.4** |
| | $\nu$ | 0.26 | 0.2652 | +2.0 | 0.2960 | +13.8 | **0.2558** | **-1.6** |
| | $v_f$ (%) | 30.0 | **29.03** | **-3.2** | 52.74 | +75.8 | 28.24 | -5.9 |
| Task 2 | $\sigma$ (S/m) | 3.8M | 2.89M | -23.8 | 3.53M | -7.1 | **3.78M** | **-0.5** |
| | $\kappa$ (W/(m · K)) | 26.0 | 20.14 | -22.5 | 24.59 | -5.4 | **26.27** | **+1.0** |
| | $E$ (MPa) | 3000.0 | 3265.8 | +8.9 | 4946.0 | +64.9 | **3021.4** | **+0.7** |
| | $\nu$ | 0.25 | 0.2560 | +2.4 | 0.2282 | -8.7 | **0.2542** | **+1.7** |
| | $v_f$ (%) | 25.0 | 32.62 | +30.5 | 31.31 | +25.2 | **26.21** | **+4.8** |

**Quantitative Validation.** Baselines exhibit significant failures in managing coupled objectives. NSGA-II struggles with multi-objective trade-offs, resulting in a +340.6% error in stiffness for Task 1 and a +64.9% overshoot in Task 2. ReAct similarly fails to map semantic intent to feasible parameters, undershooting the thermal conductivity target in Task 1 by 37.1%. In contrast, AutoMS successfully navigates the Pareto frontier, maintaining errors consistently below 6% across all dimensions ($\kappa$, $E$, $\nu$, $v_f$). For instance, in Task 2, it achieves a thermal conductivity of 26.27 W/(m · K) and a Young's modulus of 3021.4 MPa, matching targets with high precision.

**Visual Verification.** To intuitively visualize the tradeoffs across cross-physical domains, Figure 3 presents radar charts for the representative tasks. The results demonstrate a clear contrast in multi-objective alignment: AutoMS (Red) exhibits a near-perfect overlap with the target specifications, forming a balanced polygon that satisfies all constraints simultaneously. In contrast, the baselines (Blue/Orange) produce highly skewed envelopes, indicating a failure to balance conflicting properties. This graphical disparity high-

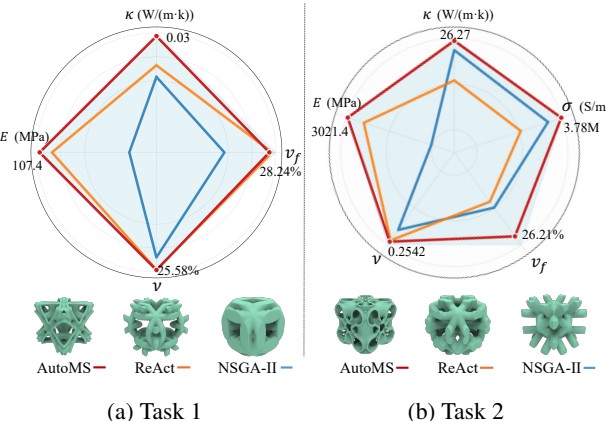

(a) Task 1        (b) Task 2

*Figure 3.* **Visual Validation of Quantitative Benchmarks.** A structural and functional comparison of the designs quantified in Table 2.

lights that while baselines may optimize single metrics, they fail to navigate the high-dimensional Pareto frontier, resulting in designs that drift significantly from the user's cross-physics intent.

## 5.3 Ablation Studies

We analyzed the Optimization Layer components to validate the neuro-symbolic synergy within AutoMS in Table 3. The table is organized into two groups: component ablations and reasoning and algorithmic engine variants.

**Impact of Core Components.** Removing the Weighted Least Squares (WLS)-based gradient estimation (the w/o Local Grad. variant) significantly degrades performance. Although it reduces average computation time (1787.4 s vs. 2180.5 s) , the Success Rate (SR) drops sharply from 83.8% to 66.7%. Without the physics-aware directional guidance provided by $\nabla f(x_k)$, the search effectively reverts to an inefficient random walk in the high-dimensional conditioning space. This is further evidenced by the decline in solution quality, with the Mean Relative Error (MRE) increasing from 0.0140 to 0.0207. These results confirm that local gradient approximation is essential for both search frequency and numerical precision

*Table 3.* **Ablation study results.** Results are organized into component ablations and reasoning and algorithmic engine variants.

| Method | Completion | | Solution Quality | | | Efficiency | |
|---|---|---|---|---|---|---|---|
| | SR ↑ | CSR ↑ | MRE ↓ | BPM ↑ | QS ↑ | Iter ↓ | Time (s) ↓ |
| **AutoMS (Full)** | **83.8%** | 59.6% | **0.0140** | **94.2%** | **82.4** | **14.4** | 2180.5 |
| *Component ablations* | | | | | | | |
| w/o Local Grad. | 66.7% | 38.7% | 0.0207 | 84.4% | 68.5 | 14.6 | **1787.4** |
| w/o Adapt. Weight | 71.7% | 26.8% | 0.0261 | 86.1% | 66.5 | 15.7 | 1965.9 |
| w/o SAES | 78.3% | **76.2%** | 0.0302 | 89.7% | 80.4 | 16.8 | 2885.9 |
| *Reasoning and algorithmic engine variants* | | | | | | | |
| w/ reasoner | 75.0% | 66.7% | 0.0154 | 89.7% | 80.7 | 18.1 | 3653.8 |
| w/o NSGA-II w/ GA | 76.9% | 61.0% | 0.0227 | 84.7% | 77.3 | 15.1 | 2767.4 |
| w/o NSGA-II w/ CMA-ES | 81.7% | 69.6% | 0.0158 | 86.9% | 81.8 | 15.2 | 2696.6 |

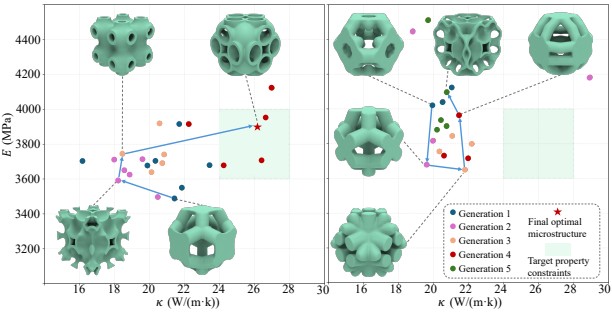

(a) AutoMS (Full)          (b) w/o SAES

*Figure 4.* **Visualization of Optimization Trajectories.** The scatter plots illustrate the evolutionary search process in the cross-physics property space ($E$ vs. $\kappa$). Candidates are color-coded by generation index to visualize temporal progression. The blue arrows trace the optimization path, highlighting the contrast between the directed convergence of AutoMS (Left) and the erratic, divergent search pattern of the baseline without SAES (Right).

**Necessity of Adaptive Weight Update (Integration).** The Adaptive Weight Update mechanism is indispensable for breaking search stagnation. The variant without this feature (w/o Adapt. Weight) yielded the lowest Constraint Satisfaction Rate at 26.8%. Without the dynamic rebalancing defined in Equation (6), the optimization frequently becomes trapped in local optima dominated by easier objectives (*e.g.*, Volume Fraction), failing to satisfy more challenging, conflicting constraints like simultaneous high stiffness and thermal conductivity. The adaptive strategy forces the system to explore the difficult regions of the Pareto frontier, ensuring that all cross-physics objectives are addressed.

**The Partial Satisfaction Trap.** Ablating the SAES module reveals a critical trade-off: the w/o SAES variant maintains a high CSR (76.2%) but a significantly lower SR (78.3%) than the full system (83.8%). This "trap" occurs because, without SAES guidance, the optimizer preferentially converges on isolated, easier objectives—inflating CSR while failing to align the coupled constraints required for full success. Figure 4 corroborates this; the ablated model stagnates on single-physics manifolds, whereas AutoMS leverages gradient updates to navigate the thermal dimension ($\kappa$) and reach the valid target zone.

**Algorithmic and Reasoner Robustness.** Replacing NSGA-II with GA or CMA-ES yields 76.9% and 81.7% SR, respectively. These results confirm that while the choice of optimization engine influences efficiency, the primary performance gains stem from the closed-loop multi-agent coordination and simulation-aware updates.

**Robustness Across LLM Logic Cores.** Table 4 shows that the framework remains stable across diverse LLM model cores. DeepSeek-V3.2 provides the best overall quality trade-off (best MRE/BPM/QS), GPT-5.2 achieves the highest SR with weaker CSR, and GPT-4o is fastest but less accurate. Qwen3-Max obtains the highest CSR, while Claude Haiku 4.5 is the most iteration-efficient. This pattern indicates that performance differences are mainly in efficiency-quality trade-offs, while the closed-loop architecture remains effective across model families.

*Table 4.* **Robustness Analysis across LLM Logic Cores.** Performance comparison of AutoMS across diverse LLM cores, showing that robustness is mainly driven by the AutoMS architecture rather than any single model.

| LLM Core | Completion | | Solution Quality | | | Efficiency | |
|---|---|---|---|---|---|---|---|
| | SR ↑ | CSR ↑ | MRE ↓ | BPM ↑ | QS ↑ | Iter ↓ | Time (s) ↓ |
| GPT-4o | 71.8% | 59.8% | 0.0439 | 84.6% | 75.5 | 13.4 | **1022.7** |
| GPT-5.2 | **84.6%** | 52.0% | 0.0188 | 87.2% | 76.2 | 12.3 | 2379.2 |
| Claude Haiku 4.5 | 76.9% | 64.7% | 0.0397 | 89.7% | 80.1 | **10.7** | 1142.0 |
| Qwen3-Max | 82.3% | **68.8%** | 0.0247 | 92.6% | 81.5 | 14.9 | 1845.8 |
| **DeepSeek-V3.2** | 83.8% | 59.6% | **0.0140** | **94.2%** | 82.4 | 14.4 | 2180.5 |

Comprehensive experimental evaluations and physical 3D-printed results are documented in Appendix C.

## 5.4 Failure Analysis

Despite AutoMS achieving a state-of-the-art success rate of 83.8%, the analysis of the remaining failure cases offers critical insight into the intrinsic topological severity of cross-physics inverse design. In high-difficulty regimes, the feasible manifold becomes sparse and disjoint due to conflicting constraints. Although our Adaptive Weight Update navigates this landscape by dynamically rebalancing gradients, unlocking regions intractable for baselines, the search must still contend with stochastic simulation feedback. Consequently, rare instances of local Pareto stagnation in coupled tri-field tasks reflect the extreme narrowness of the solution space rather than architectural limitations. These cases validate our closed-loop strategy, demonstrating its ability to probe frontiers where open-loop models collapse.

## 6 Conclusion and Future Work

We presented AutoMS, a multi-agent neuro-symbolic framework that reformulates cross-physics inverse microstructure design as an autonomous, closed-loop discovery process. By synergistically integrating LLM-driven semantic navigation with simulation-aware numerical optimization, AutoMS grounds ambiguous design intent in rigorous physical validity, effectively eliminating the "physical hallucinations" prevalent in traditional generative models. Our Simulation-Aware Evolutionary Search (SAES) mechanism enables specialized agents to navigate complex, non-differentiable landscapes by leveraging direct simulation feedback for local gradient approximation.

Empirical evaluation across 17 diverse tasks yields a state-of-the-art 83.8% success rate and a 23.3% reduction in execution time compared to existing agentic baselines. These results establish that role-specialized neuro-symbolic search offers a robust and scalable solution for inverse design problems that remain intractable for purely linguistic or numerical approaches.

**Future Work** While AutoMS sets a new standard for inverse design, we identify two strategic directions for future research. First, we aim to incorporate explicit physical laws (*e.g.*, governing PDEs) directly into the agents' reasoning logic to minimize reliance on iterative trial-and-error simulations. Second, we plan to extend the framework to multiscale design, concurrently optimizing micro-architectural unit cells and their macroscopic topological arrangements to unlock superior performance through cross-scale synergy.

## Acknowledgments

We thank the anonymous reviewers for their valuable feedback and suggestions. The work is partially supported by National Natural Science Foundation of China (No. U25A20438, 62472258, 62302275).

## Impact Statement

This paper presents work whose goal is to advance the field of machine learning. There are many potential societal consequences of our work, none of which we feel must be specifically highlighted here.

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

# A    Agent Prompts & Workflows

This appendix provides the detailed System Prompts for the core agents in the AutoMS framework, as well as the JSON Schema definitions for their callable tools and a representative interaction trace.

## A.1    System Prompts

The following System Prompts are derived directly from the agent configuration files, detailing the specific instructions, constraints, and decision logic used in the AutoMS framework.

---

**System Prompt: Manager**

**Role:** You are the conversation manager for a Microstructure Design System. Your ONLY task is to select the next speaker and provide instructions.
**Decision Rules (Priority Order):**

1. **Start:** If User input is new → Select `parser`.

2. **After Parsing:** If parser finishes → Select `generator`.

3. **After Generation:** If generator finishes → Select `simulator`.

4. **After Simulation:** Check status:
    - If "iterate" or "not satisfied" → Select `generator` (Trigger Re-generation).
    - If "satisfied" or "converged" → Select `reporter`.

5. **Termination:** After reporter → Output `TERMINATE`.

**Output Format:** You must output ONLY a valid JSON object. No other text allowed.

```
{"next_speaker": "AgentName", "instruction": "Task description..."}
```

---

**System Prompt: Parser**

**Role:** You are the Requirement Parser and Material Design Expert.
**Core Workflow:**

1. **Identify Request Type:**
    - *Type A (Application):* Use GraphRAG to map scenarios (e.g., "Heat sink") to physical properties.
    - *Type B (Explicit):* Extract numerical constraints directly.

2. **Define Physical Constraints (Stiffness):** To ensure physical validity during homogenization, strict constraints must be set:
    - Young's Modulus: $E_{target} \in [0.05, 0.40] \times E_{base}$ (for $\nu < 0.45$)
    - Consistency Check: $E = 2G(1 + \nu)$

3. **Base Material Selection:** Choose a material with sufficient stiffness to allow for the target porosity ($E_{base} \gg E_{target}$).

**Output Specification:** Return a strict JSON containing `task_type`, `specific_target` (only relevant properties), `material_parameters`, and `recommended_base_material`.

---

---

## System Prompt: Generator

**Role:** You are the Microstructure Generation Expert controlling the AI diffusion model.
**Critical Rules:**

1. **Mandatory Tool Call:** You MUST call `generate_microstructure_with_ai` in every response.

2. **Constraint Verification:** Before generating, verify strict stiffness constraints:

$$E_{target} < 0.40 \times E_{base}$$

   If targets violate this, adjust them automatically to the nearest valid value.

3. **Avoid Duplication:** Check context history. If parameters $(E, G, \nu)$ were already tried, apply a random perturbation ($\pm 5\%$) to explore new latent regions.

4. **SAES Guidance:** If `[Genetic Optimization Status]` is present, prioritize the "Evolutionary Suggested Parameters" over user initial targets.

**Workflow:** 1. Extract targets from Parser output. 2. Validate against physical limits. 3. Call generation tool. 4. Stop immediately (let Simulator take over).

---

## System Prompt: Simulator

**Role:** You are the Simulation Planner responsible for FEA validation and evolutionary strategy.
**Simulation Rules:**

1. **Base Material Physics:** Always use the *true physical properties* of the base material (e.g., Copper $\kappa = 400 \ \text{W}/(\text{m} \cdot \text{K})$ for simulation parameters, NOT the target effective properties.

2. **Iteration Logic:**

   - If any target deviation $> 10\%$, trigger **SAES Iteration**.
   - Instruction to Generator: "Adjust $E$ by $+X\%$ and re-generate."

3. **Pareto Analysis (Adaptive Multi-Objective Guidance):**

   - Identify Non-Dominated Solutions from the history pool.
   - Calculate Crowding Distance to maintain diversity.
   - Recommend "Balanced Solutions" vs. "Extreme Solutions".

**Integration:** Your output must include the `[Performance Verification Status]` tag to signal the Manager (e.g., "READY FOR FINAL EVALUATION" or "MISSING SIMULATION").

---

## System Prompt: Reporter

**Role:** You are the Final Reporter responsible for summarizing the design session.
**Reporting Requirements:**

1. **Termination:** Must end with the specific token `TERMINATE`.

2. **Pareto Report:** For multi-objective tasks, present a table of the Pareto Front:

   - Columns: Rank, ID, Objective A, Objective B.
   - Highlight the "Best Trade-off" candidate.

3. **Process Summary:** Summarize the number of iterations, the improvement in Hypervolume, and the final convergence status.

## A.2    Interaction Trace

The following is a condensed interaction trace for the task: *"Design a microstructure with precise cross-physics constraints: electrical conductivity ($\sigma = 3.8 \times 10^6$ S/m), thermal conductivity ($\kappa = 26$ W/(m · K)), Young's modulus ($E = 3$ GPa), Poisson's ratio ($\nu = 0.25$), and volume fraction ($v_f = 25\%$)."*

> **User:** "Help me design a microstructure with electrical conductivity $\sigma = 3.8 \times 10^6$ S/m, thermal conductivity $\kappa = 26$ W/(m · K), Young's modulus $E = 3$ GPa, Poisson's ratio $\nu = 0.25$, and volume fraction $v_f = 25\%$."
>
> **Manager:** Delegating to `Parser Agent`.
>
> **Parser Agent:** *Analysis:* Type B. Detected 5 simultaneous constraints spanning mechanical, thermal, and electrical domains.
> *Output:*
>
> - Base Material: Copper ($E = 110$ GPa, $\kappa = 400$ W/(m · K), $\sigma = 5.96 \times 10^7$ S/m).
> - Constraints: $E = 3000 \pm 300$ MPa, $\kappa = 26 \pm 2.6$ W/(m · K), $\sigma = (3.8 \pm 0.38) \times 10^6$ S/m, $\nu = 0.25 \pm 0.025$, $v_f = 0.25 \pm 0.025$.
> - Strategy: Cross-Physics Coupled Optimization (Full Simulation Suite Required).
>
> **Manager:** Triggering Design Prior Retrieval.
>
> **Manager:** Delegating to `Generator Agent` for innovation.
>
> **Generator Agent:** *Action:* Calling `generate_microstructure_with_ai` using Copper base.
> *Output:* Batch `copper_cross_physics_structures` (1 candidate).
>
> - Target: $E = 3000$ MPa, $G = 1200$ MPa, $\nu = 0.25$.
> - Generated: $E = 2850$ MPa ($-5.0\%$), $v_f = 24.8\%$.
>
> **Manager:** New structures detected. Triggering `Simulator`.
>
> **Simulator:** *Action:* Running `run_integrated_analysis` (stiffness + thermal + electrical) on candidates.
> *Evaluation:*
>
> - Stiffness: $E = 2847.3$ MPa (Target $E = 3000.0$ MPa, Gap $-5.1\% \rightarrow$ Acceptable).
> - Thermal: $\kappa = 22.4$ W/(m · K) (Target $\kappa = 26.0$ W/(m · K), Gap $-13.8\% \rightarrow$ Failed).
> - Electrical: $\sigma = 3.2 \times 10^6$ S/m (Target $\sigma = 3.8 \times 10^6$ S/m, Gap $-15.8\% \rightarrow$ Failed).
> - Poisson's ratio: $\nu = 0.248$ (Target 0.250, Gap $-0.8\% \rightarrow$ Acceptable).
>
> *SAES Status:* Iteration triggered (Generation $0 \rightarrow 1$).
>
> - Suggestion: Prioritize $\kappa$ and $\sigma$ jointly. Recommended adjustment: increase $v_f$ to 27% to boost conductivity paths.
>
> *Adaptive Multi-Objective Guidance Update*: Pareto analysis indicates trade-off between mechanical stiffness and transport properties; exploring higher-connectivity lattice topologies.
>
> **Manager:** Iteration 2 initiated. Triggering re-generation with SAES guidance.

# B    Implementation Details

This appendix details the specific configurations for the Simulation-Aware Evolutionary Search (SAES) algorithm, which replaces standard evolutionary operators with gradient-guided navigation to ensure reproducibility.

## B.1    SAES Configuration

The SAES framework integrates local gradient estimation (Perception), directed parameter updates (Action), and adaptive population management (Integration). The specific hyperparameters governing these phases are strictly defined in Table 5 to guarantee experimental consistency.

*Table 5.* **SAES Hyperparameters**

| Module | Parameter | Value |
|---|---|---|
| **Optimization** | Population Size ($N_{pop}$) | 20 |
| | Elite Archive Size | 50 |
| | Max Iterations | 10 |
| **Perception** | Neighbor Window ($M$) | 5 |
| (Local Gradient) | Temporal Decay ($\lambda_t$) | 0.5 |
| | Outlier Threshold | 2.5 MAD |
| **Action** | Base Step Size ($\eta$) | 0.1 |
| (Update) | Exploration Noise ($\beta$) | 0.05 |
| | Momentum Coefficient | 0.3 |
| **Integration** | Stagnation Window | 3 Generations |
| (Adaptive Weight) | Weight Range | [0.1, 2.0] |
| | Adjustment Factor ($\delta$) | +0.25 (Stagnation) |

**Integration: Adaptive Guidance Strategy.** To handle conflicting objectives (e.g., Task 6: Max $\kappa$, Min $v_f$), the scalarization weights $w_i$ are dynamically adjusted using an Adaptive Guidance strategy.

- **Conflict Detection:** If the improvement ratio $\gamma_i \approx 0$ for 3 consecutive generations, the objective is flagged as stagnant.

- **Weight Rebalancing:**

  - *Stagnation:* $w_i \leftarrow w_i \times 1.25$ (Boost priority).
  - *Fast Convergence:* $w_i \leftarrow w_i \times 0.90$ (Redistribute resources).

## B.2 Baseline Implementation

For fair comparison, baselines were implemented as follows:

- **ReAct Single Agent:** Uses the same tools but a single LLM prompt without the hierarchical Manager-Parser-Generator-Simulator structure or the SAES feedback loop.

- **NSGA-II:** A standard genetic algorithm operating on the latent space $\mathcal{X}$. It uses standard polynomial mutation and simulated binary crossover (SBX) without the gradient-guidance or adaptive weight mechanisms of SAES.

- **BayesOpt:** Standard Gaussian-process Bayesian optimization (Snoek et al., 2012) on the same conditioning variables and simulation feedback.

- **CMA-ES:** Covariance Matrix Adaptation Evolution Strategy (Hansen & Ostermeier, 2001) under the same simulation budget and stopping rule.

- **Single-Agent + SAES:** A single-agent system using the same SAES numerical update, but without role-specialized multi-agent coordination.

- **CoT + SAES:** A chain-of-thought prompting variant (Wei et al., 2022) coupled with SAES updates in a single-agent setup.

## B.3 Hybrid Retrieval-Augmented Generation

To further enhance search efficiency and mitigate the "cold start" problem in evolutionary optimization, we integrate the original MIND training dataset (comprising over 130,000 pre-computed microstructures (Xue et al., 2025)) as an offline knowledge base. The Generator agent's `generate_microstructure` tool employs a hybrid strategy that operates in two parallel streams:

- **Generative Stream (Exploration):** The diffusion model synthesizes novel geometries conditioned on the target properties $y_{target}$, exploring the continuous latent space for unseen designs.

- **Retrieval Stream (Exploitation):** Simultaneously, the tool performs search within the MIND dataset based on the requested mechanical properties.

These retrieved candidates are injected into the initial population of the Simulation-Aware Evolutionary Search (SAES) alongside the AI-generated candidates. This mechanism effectively grounds the generative process, ensuring that the evolutionary search is seeded with physically valid, high-fidelity structures while retaining the capability to innovate beyond the training distribution.

## C  Extended Experimental Evaluation and Analysis

### C.1  Detailed Performance on 17-Task Benchmark

The AutoMS framework is evaluated on a comprehensive suite of 17 diverse design tasks in Table 6. This benchmark is designed to stress-test the system across three orthogonal dimensions: optimization modality (Single vs. Multi-objective), physical complexity (Linear Elasticity vs. Non-linear Plasticity vs. Cross-physics), and constraint severity (Broad vs. Narrow Feasible Regions).

*Table 6.* **Quantitative performance analysis of the 17-task AutoMS Benchmark.** The tasks are stratified by difficulty based on the cardinality of physical constraints and simulation complexity (e.g., non-linear plasticity). Crucially, AutoMS maintains a low Mean Relative Error (MRE) across all regimes, particularly in "Hard" tasks. This demonstrates that the framework does not merely locate feasible regions but converges to high-fidelity designs that precisely match rigorous cross-physics targets.

| ID | Difficulty | Task Name | Target Objectives | MRE↓ | BPM↑ | CSR↑ | QS↑ | Iter↓ | Time(s)↓ |
|----|-----------|-----------|-------------------|------|------|------|-----|-------|----------|
| 1 | Hard | Two-Physics Precise | $\kappa = 0.035, E = 110$ MPa$, \nu = 0.26, v_f = 30\%$ | 0.0320 | 65.00% | 14.20% | 44.90 | 13.0 | 2689.2 |
| 2 | Hard | Cross-Physics Precise | $\sigma = 3.8 \times 10^6$ S/m$, \kappa = 26, E = 3$ GPa$, \nu = 0.25, v_f = 25\%$ | 0.0174 | 80.00% | 21.40% | 53.09 | 12.9 | 2034.1 |
| 3 | Easy | High Electrical Cond. | Max $\sigma$ ($> 3.5 \times 10^6$ S/m) | 0.0010 | 100.00% | 38.40% | 81.51 | 11.2 | 1535.4 |
| 4 | Medium | Thermal-Stiffness | $\kappa \approx 26, E \approx 3.8$ GPa | 0.0253 | 100.00% | 62.60% | 88.53 | 17.0 | 2975.9 |
| 5 | Medium | Tri-Field Coupling | $\kappa \approx 25, \sigma \approx 3 \times 10^6$ S/m$, E \approx 3$ GPa | 0.0195 | 100.00% | 67.50% | 90.06 | 18.8 | 2693.3 |
| 6 | Medium | Pareto Thermal-Mass | Max $\kappa$, Min $v_f$ | 0.0072 | 100.00% | 78.70% | 93.54 | 15.5 | 2153.8 |
| 7 | Medium | Three-Obj Pareto | Max $E, \kappa$, Min $v_f$ | 0.0036 | 100.00% | 79.40% | 93.78 | 16.2 | 2096.4 |
| 8 | Medium | Iterative Design | $E \to 4$ GPa$, \kappa \to 25$ | 0.0010 | 100.00% | 96.20% | 98.85 | 9.0 | 876.2 |
| 9 | Hard | Material Constraint | Ti6Al4V: $\kappa \approx 1.2, E \approx 6$ GPa | 0.0414 | 100.00% | 69.10% | 85.32 | 20.0 | 2870.7 |
| 10 | Medium | Extreme Performance | $\kappa > 35, E > 4$ GPa$, v_f \in [0.3, 0.4]$ | 0.0043 | 100.00% | 51.60% | 85.44 | 10.8 | 1501.2 |
| 11 | Hard | Flexible Polymer | Max $E$ ($> 400$ MPa)$, v_f < 0.50$ | 0.0029 | 87.50% | 87.20% | 86.13 | 10.2 | 1561.1 |
| 12 | Medium | Al6061 Pareto | Max $\kappa$, Min $v_f$ | 0.0035 | 100.00% | 65.50% | 89.62 | 15.2 | 2367.2 |
| 13 | Medium | AlSi10Mg Pareto | Max $E, \kappa$, Min $v_f$ | 0.0046 | 100.00% | 75.60% | 92.63 | 20.2 | 2768.4 |
| 14 | Hard | Thermal-Plasticity | $\kappa \approx 20, E \approx 3.5$ GPa, Plasticity | 0.0178 | 85.00% | 25.40% | 66.44 | 14.5 | 2257.6 |
| 15 | Hard | High Strength | Max $E$ ($> 10$ GPa), Plasticity | 0.0152 | 85.00% | 34.20% | 64.11 | 14.5 | 2158.4 |
| 16 | Easy | High Thermal Cond. | Max $\kappa$ ($> 30$)$, v_f \in [0.2, 0.3]$ | 0.0157 | 100.00% | 47.30% | 84.03 | 12.8 | 1884.0 |
| 17 | Easy | High Stiffness | Max $E$ ($> 5$ GPa)$, v_f < 0.25$ | 0.0112 | 100.00% | 99.10% | 99.62 | 12.0 | 2645.8 |

- **Regime I: Fundamental Validation & Single-Physics (Tasks 3-4, 8, 16-17)** These tasks establish the baseline capabilities of the framework in optimizing single or dual physical properties. AutoMS demonstrates exceptional precision in this regime. Notably, in Task 3 (High Electrical Conductivity), the system achieves a negligible MRE of 0.001 (0.1%), demonstrating that the Local Gradient Approximation (Perception Module) can effectively fine-tune microstructures to near-machine-precision levels. By minimizing the deviation between the target $\sigma$ and the simulated properties to virtually zero, AutoMS proves that LLM-driven evolutionary search can overcome the stochasticity typically associated with generative models.

- **Regime II: Multi-Objective Pareto & Tri-Field Coupling (Tasks 5-7, 10, 12-13)** This regime tests the agent's ability to resolve conflicting objectives and handle coupled fields. For Task 6 (Pareto Thermal-Mass), AutoMS achieves a remarkably low MRE of 0.0072, alongside a high Quality Score (QS) of 93.54. This indicates that the Adaptive Weight Update mechanism successfully prevents the search from collapsing into trivial solutions (e.g., solid blocks). Instead of merely finding a "feasible" compromise, the system converges accurately onto the high-performance non-dominated front, balancing thermal conductivity and volume fraction with high fidelity.

- **Regime III: High-Complexity Constraints & Non-Linearity (Tasks 1-2, 9, 11, 14-15)** Classified as "Hard" in our benchmark, these tasks represent the most challenging edge cases, including High-Dimensional Precision Targeting and Non-Linear Plasticity. In Task 9 (Material Constraint: Ti6Al4V), where the feasible manifold is disjoint and sparse, traditional baselines often exhibit significant drift. In contrast, AutoMS limits the MRE to 0.0414 (approx. 4%),

maintaining tight adherence to the material constraints. Furthermore, in Task 1 (Two-Physics Precise), the system achieves an MRE of 0.0320, proving that even under five simultaneous constraints, the collaborative agent framework effectively navigates the landscape to locate the precise intersection of physical validity.

In Figure 5, we visualize the optimal microstructures generated for each task, revealing how AutoMS autonomously adapts topological features to satisfy cross-physics constraints.

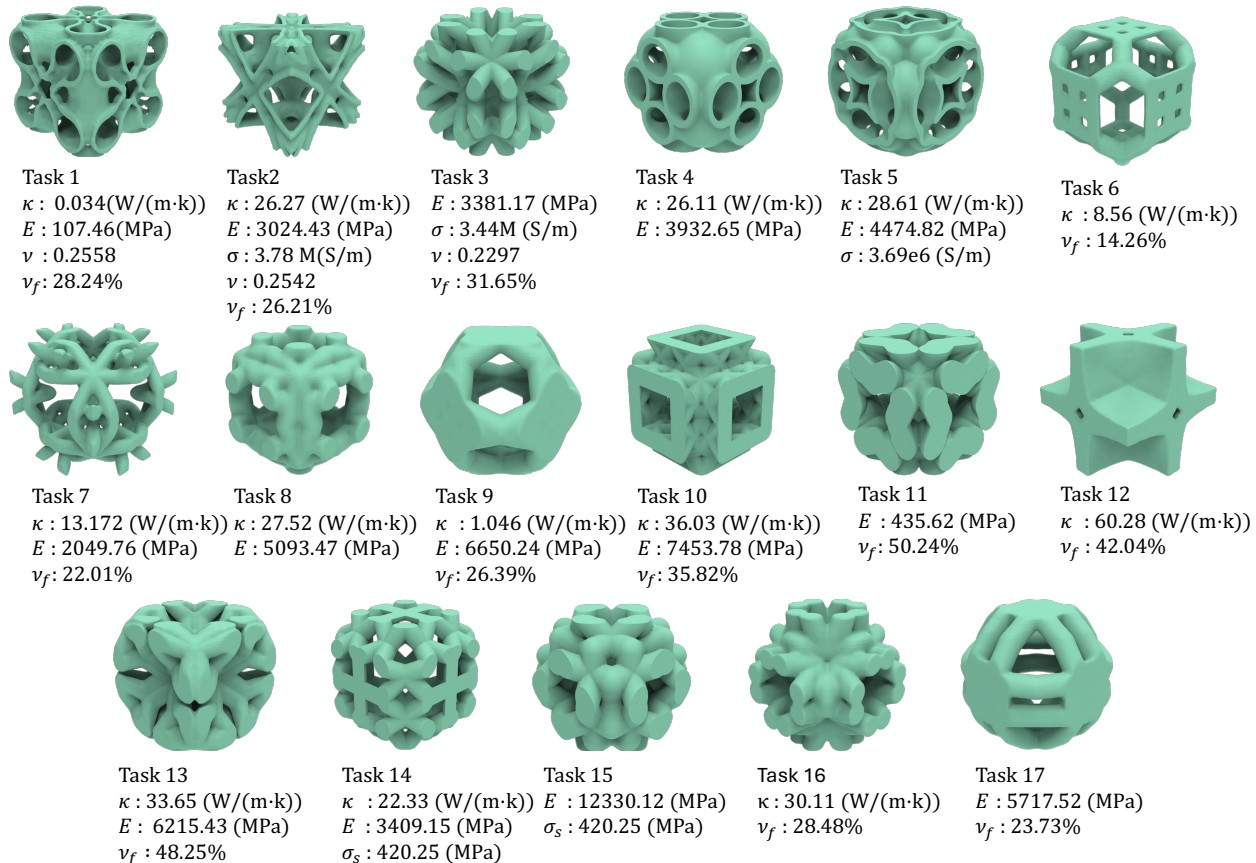

Task 1
$\kappa$ : 0.034(W/(m·k))
$E$ : 107.46(MPa)
$\nu$ : 0.2558
$\nu_f$: 28.24%

Task2
$\kappa$ : 26.27 (W/(m·k))
$E$ : 3024.43 (MPa)
$\sigma$ : 3.78 M(S/m)
$\nu$ : 0.2542
$\nu_f$ : 26.21%

Task 3
$E$ : 3381.17 (MPa)
$\sigma$ : 3.44M (S/m)
$\nu$ : 0.2297
$\nu_f$ : 31.65%

Task 4
$\kappa$ : 26.11 (W/(m·k))
$E$ : 3932.65 (MPa)

Task 5
$\kappa$ : 28.61 (W/(m·k))
$E$ : 4474.82 (MPa)
$\sigma$ : 3.69e6 (S/m)

Task 6
$\kappa$ : 8.56 (W/(m·k))
$\nu_f$ : 14.26%

Task 7
$\kappa$ : 13.172 (W/(m·k))
$E$ : 2049.76 (MPa)
$\nu_f$: 22.01%

Task 8
$\kappa$ : 27.52 (W/(m·k))
$E$ : 5093.47 (MPa)

Task 9
$\kappa$ : 1.046 (W/(m·k))
$E$ : 6650.24 (MPa)
$\nu_f$: 26.39%

Task 10
$\kappa$ : 36.03 (W/(m·k))
$E$ : 7453.78 (MPa)
$\nu_f$: 35.82%

Task 11
$E$ : 435.62 (MPa)
$\nu_f$: 50.24%

Task 12
$\kappa$ : 60.28 (W/(m·k))
$\nu_f$ : 42.04%

Task 13
$\kappa$ : 33.65 (W/(m·k))
$E$ : 6215.43 (MPa)
$\nu_f$ : 48.25%

Task 14
$\kappa$ : 22.33 (W/(m·k))
$E$ : 3409.15 (MPa)
$\sigma_s$ : 420.25 (MPa)

Task 15
$E$ : 12330.12 (MPa)
$\sigma_s$ : 420.25 (MPa)

Task 16
$\kappa$ : 30.11 (W/(m·k))
$\nu_f$ : 28.48%

Task 17
$E$ : 5717.52 (MPa)
$\nu_f$ : 23.73%

*Figure 5.* **Gallery of Optimized Microstructures.** A visualization of the best-performing unit cell for each of the 17 benchmark tasks. The diversity of the generated topologies highlights AutoMS's ability to discover physics-aware geometries

**Physical Realization.** To verify the practical utility of our neuro-symbolic discovery process, we transitioned the digital designs to physical prototypes as shown in Figure 6. We fabricated representative microstructures using high-precision 3D printing across diverse task topologies. The bottom panels of Figure 6 demonstrate that the complex, optimized features are robustly manufacturable. This alignment underscores the reliability of the AutoMS framework in ensuring that inverse-designed structures are not merely digital artifacts but are physically valid and predictable.

## C.2   One-shot Generator Baseline

To quantify the intrinsic generative capability of the MIND backbone and justify the necessity of our iterative multi-agent coordination, we evaluate a One-shot MIND Baseline. In this configuration, the generator receives each target specification once and produces a single microstructure geometry without the benefit of SAES updates, agentic correction, or closed-loop feedback.

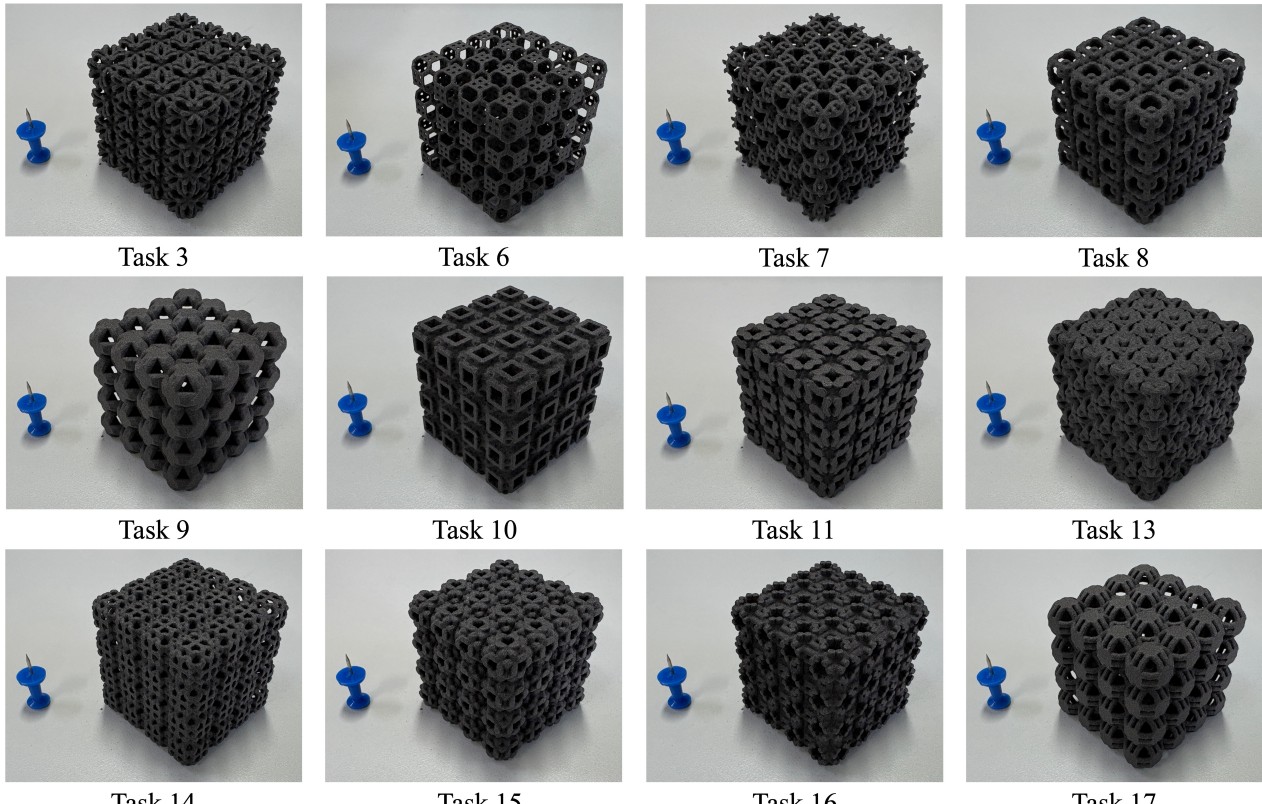

*Figure 6.* **Gallery of 3D-printed microstructures.** These representative samples showcase the topological diversity and robust manufacturability of the optimal designs discovered by AutoMS across the 17-task benchmark.

**Quantitative Performance and Difficulty Sensitivity.** As summarized in Table 7, the one-shot baseline succeeds in only 6 out of 17 tasks (35.3% success rate). The performance exhibits a stark sensitivity to task difficulty:

- **Easy Regimes**: In single-objective or loosely constrained tasks (e.g., T3, T6, T7), the one-shot model achieves a 100% Success Rate with negligible Mean Relative Error (MRE).

- **Hard Regimes**: In tasks involving tightly coupled cross-physics constraints (e.g., T1, T2, T11), the one-shot SR collapses to 0%. For T1 and T2, the MRE exceeds 0.60, indicating a total failure to locate the valid physical manifold in high-dimensional conditioning space.

**Addressing Physical Hallucinations.** These results provide empirical evidence of the physical hallucinations inherent in one-shot generative mappings. While the generator can synthesize visually plausible topologies, it lacks the internal reasoning to navigate conflicting physical objectives (e.g., simultaneous high stiffness and thermal conductivity) without external guidance.

**The Role of the Simulator.** Crucially, this baseline clarifies that the cross-physics simulator acts strictly as an evaluative oracle, not a repair mechanism. It identifies topological or physical invalidity but cannot smooth over or correct a hallucinated generation. Consequently, the performance gap between the one-shot baseline and the full AutoMS framework (35.3% vs. 83.8% SR) is strictly attributable to the closed-loop SAES mechanism, which leverages simulation feedback to iteratively steer the search toward physically valid Pareto frontiers.

*Table 7.* **One-shot MIND baseline on the 17-task suite.** This baseline performs single-pass generation without iterative SAES correction.

| Task | Difficulty | SR | CSR | MRE | BPM | QS |
|------|-----------|------|------|--------|------|-------|
| T1 | Hard | 0% | 0% | 0.6099 | 33% | 17.2 |
| T2 | Hard | 0% | 0% | 0.6067 | 25% | 13.9 |
| T3 | Easy | 100% | 100% | 0.0010 | 100% | 100.0 |
| T4 | Medium | 0% | 0% | 0.0375 | 33% | 23.0 |
| T5 | Medium | 0% | 80% | 0.0452 | 75% | 63.5 |
| T6 | Medium | 100% | 100% | 0.0048 | 100% | 100.0 |
| T7 | Medium | 100% | 100% | 0.0035 | 100% | 100.0 |
| T8 | Medium | 100% | 80% | 0.0010 | 100% | 94.0 |
| T9 | Hard | 0% | 75% | 0.3289 | 67% | 55.9 |
| T10 | Medium | 0% | 0% | 0.0920 | 33% | 22.4 |
| T11 | Hard | 0% | 0% | 1.0000 | 0% | 0.0 |
| T12 | Medium | 100% | 100% | 0.0048 | 100% | 100.0 |
| T13 | Medium | 100% | 100% | 0.0035 | 100% | 100.0 |
| T14 | Hard | 0% | 0% | 0.4406 | 25% | 15.6 |
| T15 | Hard | 0% | 0% | 0.0974 | 33% | 22.4 |
| T16 | Easy | 0% | 0% | 0.8542 | 0% | 1.5 |
| T17 | Easy | 0% | 25% | 0.0985 | 50% | 36.5 |

## C.3   Statistical Stability Analysis

*Table 8.* **Statistical robustness over four independent runs.** Mean and standard deviation are reported on the 17-task benchmark.

| Metric | Mean | Standard Deviation |
|--------|--------|--------------------|
| SR | 83.8% | 3.3% |
| CSR | 59.6% | 13.1% |
| MRE | 0.0140 | 0.0072 |
| BPM | 94.2% | 1.7% |
| QS | 82.4 | 4.67 |
| Iter | 14.4 | 8.05 |
| Time (s) | 2180.5 | 1346.13 |

In this section, we quantify the statistical robustness of the AutoMS framework across the 17-task benchmark suite. To ensure reliable results, all headline performance metrics are aggregated over four independent runs per task to account for the stochastic nature of LLM inference and the diffusion-based generative process.

**Stability of Core Success Metrics.**   As reported in Table 8, AutoMS demonstrates high algorithmic stability. The Success Rate (SR) maintains a low standard deviation of 3.3% around a mean of 83.8%, while the Best Property Match (BPM) exhibits an even lower variance of 1.7%. These results indicate that the framework consistently converges to the optimal physical manifolds across repeated trials, effectively mitigating the randomness inherent in open-loop neural models through its closed-loop feedback mechanism.

**Efficiency and Difficulty Stratification.**   In contrast, metrics related to computational effort—namely Wall-clock Time (SD = 1346.13s) and Agent Iterations (SD = 8.05), show higher variability. This variance is not a reflection of algorithmic instability but rather a direct result of the difficulty stratification across the 17 tasks. While "Easy" tasks often reach convergence within a few iterations, "Hard" tasks involving coupled tri-field physics or non-linear plasticity require significantly more search rounds and simulation cycles to achieve full validity.Robustness of Intermediate States. The Constraint Satisfaction Rate (CSR) shows a standard deviation of 13.1%. This variability reflects the sensitivity of initial search trajectories to stochastic perturbations. However, the stability of the final SR suggests that the Simulation-Aware Evolutionary Search (SAES) mechanism reliably steers these diverse intermediate candidates toward the rigorous target specifications.

Overall, this statistical analysis confirms that AutoMS provides a reproducible and robust pathway for autonomous materials discovery, meeting the high standards of reliability required for scientific applications.

## C.4 Sensitivity to Generator Backbone

We further investigated the framework's dependence on the generative model by replacing the high-fidelity MIND (Xue et al., 2025) generator with a comparatively weaker voxel-based diffusion model from (Yang et al., 2026).

As detailed in Table 9, utilizing a weaker generator leads to a performance drop in SR (83.8% to 73.3%) and BPM (94.2% to 88.9%), alongside a significant increase in wall-clock time from 2180.5s to 4223.3s. This increase in runtime suggests that lower-quality initial designs require more SAES iterations to achieve physical validity.

Crucially, even with a significantly weaker generator, AutoMS still outperforms standalone numerical baselines such as Bayesian Optimization (65.0% SR) and CMA-ES (66.7% SR). This demonstrates that the performance advantages of AutoMS are not solely inherited from the generative backbone but are fundamentally driven by the framework's ability to utilize simulation feedback to steer even suboptimal candidates toward the target physical manifold.

*Table 9.* **Generator dependence analysis.** MIND is replaced by the weaker generator from Yang et al. (Yang et al., 2026) while keeping the AutoMS framework unchanged.

| Generator | SR ↑ | CSR ↑ | MRE ↓ | BPM ↑ | QS ↑ | Iter ↓ | Time (s) ↓ |
|---|---|---|---|---|---|---|---|
| MIND (Xue et al., 2025) | 83.8% | 59.6% | 0.0140 | 94.2% | 82.4 | 14.4 | 2180.5 |
| Yang et al. (Yang et al., 2026) | 73.3% | 57.7% | 0.0137 | 88.9% | 80.7 | 12.3 | 4223.3 |

## C.5 Compute and Token Budget.

To quantify the scalability of AutoMS, we profile the runtime and token consumption across the 17-task benchmark suite. As detailed in Table 10, the execution cost is dominated by high-fidelity physics simulations and tool executions, which account for 69.2% (1508.9 s) of the average 2180.5 s total wall-clock time per task. LLM inference accounts for the remaining 30.7% (669.4 s).

The average token usage is 457306 per task, comprising 436911 input tokens and 20395 output tokens. Combined with the runtime breakdown, this distribution indicates that the system's efficiency gains are primarily driven by the orchestration layer's capacity to filter physically invalid requests and prune the search space prior to engaging the expensive simulation engine. By strategically managing these interactions, the orchestration layer prevents redundant simulation cycles, ensuring that computational resources are focused on the most promising design candidates.

*Table 10.* **Compute and token breakdown per task.** Statistics are averaged over four runs on the 17-task benchmark.

| Metric | Value |
|---|---|
| Total wall-clock time | 2180.5 s |
| LLM inference time | 669.4 s (30.7%) |
| Simulation / tool execution time | 1508.9 s (69.2%) |
| Average token usage | 457,306 |
| Input tokens | 436,911 |
| Output tokens | 20,395 |

# D Physics Simulation

This section provides the detailed mathematical formulations for the homogenization and plasticity simulations used in the AutoMS framework.

## D.1 Homogenization Theory

We consider a periodic Representative Volume Element (RVE) $\Omega$ with domain size $L$ (Dong et al., 2019; Xing et al., 2025). The total strain field is decomposed into macroscopic and fluctuation components:

$$\epsilon_{ij}(\mathbf{x}) = \bar{\epsilon}_{ij} + \tilde{\epsilon}_{ij}(\mathbf{x}), \tag{7}$$

where $\bar{\epsilon}_{ij}$ is the prescribed macroscopic strain and $\tilde{\epsilon}_{ij}$ is the periodic fluctuation field satisfying $\langle \tilde{\epsilon}_{ij} \rangle = 0$.

The homogenized elastic tensor $C^H \in \mathbb{R}^{6 \times 6}$ relates macroscopic stress and strain:

$$\overline{\sigma}_{ij} = C_{ijkl}^H \overline{\epsilon}_{kl}. \tag{8}$$

For each of the six independent macroscopic strain cases $\overline{\epsilon}^{(m)}$ ($m = 1, \ldots, 6$), the displacement field $u^{(m)}$ is obtained by solving the weak form:

$$\int_\Omega C_{ijpq}^b \epsilon_{ij}(v) \epsilon_{pq}(u^{(m)}) \, \mathrm{d}\Omega = \int_\Omega C_{ijpq}^b \epsilon_{ij}(v) \overline{\epsilon}_{pq}^{(m)} \, \mathrm{d}\Omega, \quad \forall v \in \mathcal{V}, \tag{9}$$

where $C^b$ is the base material stiffness tensor and $\mathcal{V}$ is the space of periodic test functions.

**Periodic Boundary Conditions.** To rigorously compute the effective properties, we treat the RVE as a building block that repeats infinitely in all directions. Following the standard computational homogenization approach (Dong et al., 2019), we apply Periodic Boundary Conditions (PBCs) to ensure deformation continuity across adjacent unit cells.

The fundamental idea is to decompose the total displacement field $\boldsymbol{u}(\boldsymbol{x})$ into two components:

$$u_i(\boldsymbol{x}) = \underbrace{\overline{\epsilon}_{ij} x_j}_{\text{Macroscopic Linear}} + \underbrace{\tilde{u}_i(\boldsymbol{x})}_{\text{Periodic Fluctuation}}, \tag{10}$$

where $\overline{\epsilon}_{ij}$ represents the average macroscopic strain applied to the material, and $\tilde{u}_i(\boldsymbol{x})$ captures the local non-linear deformations caused by the complex microstructure geometry.

Crucially, the fluctuation term $\tilde{u}_i$ must be identical on opposite boundaries to maintain periodicity. This leads to the kinematic constraint enforced on the simulation: for every pair of corresponding nodes $\boldsymbol{x}^+$ and $\boldsymbol{x}^-$ on opposite RVE faces (including edges and corners):

$$u_i(\boldsymbol{x}^+) - u_i(\boldsymbol{x}^-) = \overline{\epsilon}_{ij}(x_j^+ - x_j^-). \tag{11}$$

In our finite element implementation, this constraint links the degrees of freedom of paired nodes, ensuring that the RVE deforms cooperatively with its neighbors under the prescribed macroscopic strain (Dong et al., 2019).

**Elastic Tensor Computation.** The components of $C^H$ are computed by summing the contributions of all active voxels:

$$C_{mnpq}^H = \frac{1}{|\Omega|} \int_\Omega \left( \overline{\epsilon}_{ij}^{(m)} - \epsilon_{ij}(u^{(m)}) \right) C_{ijkl}^b \left( \overline{\epsilon}_{kl}^{(p)} - \epsilon_{kl}(u^{(p)}) \right) \, \mathrm{d}\Omega. \tag{12}$$

**Engineering Constants.** From the compliance tensor $S^H = (C^H)^{-1}$, the effective engineering constants are:

$$E_x = \frac{1}{S_{11}^H} \qquad E_y = \frac{1}{S_{22}^H} \qquad E_z = \frac{1}{S_{33}^H} \tag{13}$$

$$G_{xy} = \frac{1}{S_{66}^H} \qquad G_{xz} = \frac{1}{S_{55}^H} \qquad G_{yz} = \frac{1}{S_{44}^H} \tag{14}$$

$$\nu_{xy} = -\frac{S_{12}^H}{S_{11}^H} \qquad \nu_{xz} = -\frac{S_{13}^H}{S_{11}^H} \qquad \nu_{yz} = -\frac{S_{23}^H}{S_{22}^H} \tag{15}$$

For isotropic or near-isotropic structures, the averaged Young's modulus $E$, shear modulus $G$, and Poisson's ratio $\nu$ are computed as:

$$E = \frac{E_x + E_y + E_z}{3}, \quad G = \frac{G_{xy} + G_{xz} + G_{yz}}{3}, \quad \nu = \frac{\nu_{xy} + \nu_{xz} + \nu_{yz}}{3}. \tag{16}$$

### D.2 Plasticity Simulation

For nonlinear material response, we employ the J2 (von Mises) plasticity framework with isotropic hardening.

**Stress Decomposition.** The Cauchy stress tensor $\sigma_{ij}$ is decomposed into hydrostatic and deviatoric parts:

$$\sigma_{ij} = \underbrace{\frac{1}{3}\sigma_{kk}\delta_{ij}}_{\text{hydrostatic}} + \underbrace{s_{ij}}_{\text{deviatoric}}, \tag{17}$$

where $s_{ij} = \sigma_{ij} - \frac{1}{3}\sigma_{kk}\delta_{ij}$ is the deviatoric stress tensor.

**Yield Function.** The von Mises yield criterion is defined as:

$$f(\sigma, \bar{\epsilon}^p) = \sigma_{vm} - \sigma_y(\bar{\epsilon}^p) = \sqrt{\frac{3}{2}s_{ij}s_{ij}} - \sigma_y(\bar{\epsilon}^p) \leq 0, \tag{18}$$

where $\sigma_{vm}$ is the von Mises equivalent stress, $\sigma_y$ is the current yield stress, and $\bar{\epsilon}^p$ is the equivalent plastic strain.

**Flow Rule.** The plastic strain rate is given by the associative flow rule:

$$\dot{\epsilon}_{ij}^p = \dot{\gamma}\frac{\partial f}{\partial \sigma_{ij}} = \dot{\gamma}\frac{3s_{ij}}{2\sigma_{vm}}, \tag{19}$$

where $\dot{\gamma} \geq 0$ is the plastic multiplier satisfying the Kuhn-Tucker conditions:

$$\dot{\gamma} \geq 0, \quad f \leq 0, \quad \dot{\gamma}f = 0. \tag{20}$$

**Isotropic Hardening.** The yield stress evolution follows the Swift hardening law:

$$\sigma_y(\bar{\epsilon}^p) = \sigma_{y0}\left(1 + \frac{\bar{\epsilon}^p}{\epsilon_0}\right)^n, \tag{21}$$

where $\sigma_{y0}$ is the initial yield stress, $\epsilon_0$ is a reference strain, and $n$ is the hardening exponent.

**Return Mapping Algorithm.** For implicit time integration, the stress update is performed using the radial return algorithm:

1. Compute trial elastic stress: $\sigma_{ij}^{\text{trial}} = \sigma_{ij}^n + C_{ijkl}^b\Delta\epsilon_{kl}$

2. Check yield condition: If $f(\sigma^{\text{trial}}) \leq 0$, accept elastic step ($\sigma_{ij}^{n+1} = \sigma_{ij}^{\text{trial}}$).

3. Otherwise, solve for $\Delta\gamma$ via consistency:

$$\sigma_{vm}^{n+1} = \sigma_{vm}^{\text{trial}} - 3G\Delta\gamma = \sigma_y(\bar{\epsilon}_n^p + \Delta\gamma) \tag{22}$$

4. Update deviatoric stress:

$$s_{ij}^{n+1} = \left(1 - \frac{3G\Delta\gamma}{\sigma_{vm}^{\text{trial}}}\right)s_{ij}^{\text{trial}} \tag{23}$$

5. Reconstruct the final stress tensor (preserving the elastic hydrostatic part):

$$\sigma_{ij}^{n+1} = s_{ij}^{n+1} + \frac{1}{3}\sigma_{kk}^{\text{trial}}\delta_{ij} \tag{24}$$

**Energy Absorption.** The total plastic work (energy absorption) is computed as:

$$W_p = \int_0^{\bar{\epsilon}^p} \sigma_y(\xi)\,d\xi. \tag{25}$$

Due to the significant computational cost of nonlinear plasticity simulations, their execution time is excluded from the overall efficiency statistics reported in the experiments.

## D.3 Simulation Parameters

- **Grid Resolution:** $64^3$ voxels.

- **Element Type:** 8-node Hexahedral (C3D8) with reduced integration.

- **Solver:** Preconditioned Conjugate Gradient (PCG).

- **Convergence:** Residual norm $< 10^{-6}$.

- **Plasticity Tolerance:** $|\Delta\gamma - \Delta\gamma_{\text{prev}}| < 10^{-8}$.

# E   Evaluation Metrics Definition

To rigorously quantify the performance of AutoMS, we employ a composite set of metrics covering task completion, solution quality, and efficiency. The precise definitions and calculation methods are detailed below.

## E.1   Task Completion Metrics

**Success Rate (SR).**   SR measures the reliability of the system in finding at least one fully valid solution. A run is considered successful if there exists at least one generated microstructure $x$ in the candidate set $\mathcal{X}$ that satisfies all $K$ property targets within the tolerance threshold $\delta$ (set to 10%):

$$\text{SR} = \frac{1}{R}\sum_{r=1}^{R}\mathbb{I}(\exists x \in \mathcal{X}_r : \forall j \in \{1,\ldots,K\}, |P_j(x) - T_j| \leq \delta \cdot T_j), \tag{26}$$

where $R$ is the total number of independent runs, $P_j(x)$ is the $j$-th property value of microstructure $x$, and $T_j$ is the target value.

**Constraint Satisfaction Rate (CSR).**   CSR evaluates the consistency of the agent in generating "partially correct" designs. It is defined as the proportion of generated candidates that satisfy at least 50% of the design objectives:

$$\text{CSR} = \frac{1}{|\mathcal{X}_{total}|}\sum_{x \in \mathcal{X}_{total}}\mathbb{I}\left(\sum_{j=1}^{K}\mathbb{I}(|P_j(x) - T_j| \leq \delta \cdot T_j) \geq 0.5K\right). \tag{27}$$

A higher CSR indicates that the agent effectively navigates the search space towards the target region, even if some candidates fail strict validation.

## E.2   Solution Quality Metrics

**Mean Relative Error (MRE).**   MRE quantifies the precision of the *best* candidate found in a run. For a run $r$, let $x^*$ be the candidate with the minimum average relative error. The reported MRE is averaged across all runs:

$$\text{MRE} = \frac{1}{R}\sum_{r=1}^{R}\min_{x \in \mathcal{X}_r}\left(\frac{1}{K}\sum_{j=1}^{K}\frac{|P_j(x) - T_j|}{T_j}\right). \tag{28}$$

**Relative Error (Err).** While Mean Relative Error (MRE) quantifies the magnitude of deviation, the Relative Error (Err) measures the signed directional deviation of a specific physical property from its target. This metric is used in Table 2 to distinguish between undershooting (negative) and overshooting (positive) target specifications, calculated as:

$$\text{Err}(x) = \frac{P(x) - T}{T} \times 100\%, \tag{29}$$

where $P(x)$ denotes the simulated value of the property for candidate $x$, and $T$ represents the prescribed target value. A positive value indicates the property exceeds the target, while a negative value indicates a deficiency.

**Best Property Match (BPM).** BPM measures the maximum number of objectives simultaneously satisfied by a single candidate, expressed as a percentage. It reflects the peak capability of the system to handle multi-objective conflicts:

$$\text{BPM} = \frac{1}{R} \sum_{r=1}^{R} \max_{x \in \mathcal{X}_r} \left( \frac{1}{K} \sum_{j=1}^{K} \mathbb{I}(|P_j(x) - T_j| \leq \delta \cdot T_j) \right) \times 100\%. \tag{30}$$

**Quality Score (QS).** To provide a holistic ranking of different methods, we introduce a composite Quality Score (0-100) that aggregates success status, constraint satisfaction, and precision. It is calculated as:

$$\text{QS} = 20 \cdot \mathbb{I}_{success} + 40 \cdot \text{BPM} + 30 \cdot \text{CSR} + 10 \cdot (1 - \min(\text{MRE}, 1.0)). \tag{31}$$

The weights (20, 40, 30, 10) are empirically assigned to prioritize property matching (BPM) and consistency (CSR), while rewarding final success and precision.

### E.3 Efficiency Metrics

**Agent Iterations (Iter).** A method-specific convergence-cycle proxy. For agentic methods, Iter counts major interaction/search rounds; for evolutionary baselines, it counts generations. We report the mean Iter over four independent runs, so decimal values are expected. This excludes internal inference steps and counts only the major feedback loops between the Generator and Simulator.

**Wall-clock Time.** The total execution time in seconds, including LLM inference, database retrieval, and cross-physics simulation. Note that for plasticity tasks, the simulation time is significantly longer and is reported separately or excluded from aggregate efficiency comparisons to avoid skewing the results.

