# OpenReview forum: "AutoMS: Multi-Agent Evolutionary Search for Cross-Physics Inverse Microstructure Design"
_ICML.cc/2026/Conference — ICML 2026 regular_

### Official Review · Reviewer_5yWc · 2026-03-12

**Soundness:** 3
**Presentation:** 3
**Significance:** 2
**Originality:** 3
**Overall Recommendation:** 4
**Confidence:** 3

**Summary:**

AutoMS designs the microstructure of materials through agents and has achieved good results on 17 physical tasks. However, there are still some issues in this paper that can be optimized.

**Compliance With Llm Reviewing Policy:**

Affirmed.

**Final Justification:**

This paper uses a multi-agent system to automatically design microstructures. I believe it is a valuable paper and hope it will be accepted.

**Key Questions For Authors:**

1. There are obvious grammatical and formatting issues in the manuscript. The authors need to carefully proofread the paper. For example, in the abstract, ”xx”.

2. The comparison algorithms in this paper include only NSGA-II and ReAct, and no more recent algorithms are considered. In addition, is NSGA-II the only multi-objective algorithm in this field? Finally, the paper only compares with algorithms from the computer science domain. .Have the authors compared the results with the performance of real materials?

3. During the microstructure design stage, how is the search space defined? How do the authors ensure that the designed material architectures conform to physical laws?

4. Can AutoMS truly design samples with performance superior to existing materials, or is the work limited to simple simulations?

5. The LLM appears to be mainly applied in the Orchestration layer. The optimization layer does not seem to involve LLMs. Is this because the main difficulty of this scientific task lies in semantic understanding?

**Limitations:**

yes

**Strengths And Weaknesses:**

Strengths:

The AutoMS method proposed in this paper is valuable and interesting. The approach uses a multi-agent system to automatically design microstructures for high-performance materials, which is meaningful. In addition, the method considers a multi-objective optimization problem, which is important.

Weaknesses:

Although the AutoMS method is valuable, there are several issues regarding the completeness of the manuscript.
The experimental section only compares the method with NSGA-II and ReAct, lacking comparisons with more algorithms, especially more advanced ones.
In the microstructure design process, the paper does not clearly explain how the search space is defined.
Moreover, the experiments in the paper appear to be mainly conducted in a simulation environment.
It remains unclear whether AutoMS can design real material structures whose performance surpasses that of existing materials, or whether it only produces designs that satisfy the required conditions within a simulated environment.

I would be happy to change my score if the authors can address some of the issues.

---

> ### Author Rebuttal · Authors · 2026-03-31
>
> Thank you for your thoughtful review and insightful questions. We appreciate your constructive suggestions on baselines, search space, and real-world validation. To address these points, we have conducted additional experiments and will incorporate the results in the revision.
>
> **Q1. Sim-to-Real Validation via Fabrication and Physical Testing (Key Questions 2 & 4)**
>
> 1. Manufacturability & Sim-to-Real Agreement:
>
> We **3D-printed** AutoMS-designed microstructures **(https://anonymous.4open.science/r/ICML_2026_AutoMS-4763/print.jpg)** and performed **uniaxial compression tests** **(https://anonymous.4open.science/r/ICML_2026_AutoMS-4763/phy.jpg)**. The physical measurements (solid lines) show strong quantitative agreement with our FEM predictions (dashed lines). This confirms our designs are not merely digital artifacts, but are physically manufacturable and accurately simulated.
>
> 2. Superiority Over Existing Materials:
>
> We clarify that "surpassing existing materials" does not mean exceeding the absolute stiffness of the solid base material, which is physically constrained. Instead, it refers to outperforming standard existing microstructures (e.g., Gyroid, Octet-truss) at the same relative density. Specifically, AutoMS designs achieve superior coupled cross-physics objectives, such as maximizing structural stiffness while maintaining target thermal and structural stiffness properties in Task 1.
>
>
> **Q2. Broader Baseline Comparisons (Weakness 2 & Key Question 2)**
> We agree that broader baseline comparisons are important. To address this, we added several stronger optimizers and agent variants. Among stand-alone numerical methods, Bayesian Optimization achieves 65.0% SR and CMA-ES achieves 66.7% SR. Within our framework, replacing SAES with other backends gives 76.9% SR with GA and 81.7% SR with CMA-ES. By comparison, the full AutoMS (with SAES) achieves the best overall performance, with 83.8% SR, 0.0140 MRE, 94.2% BPM, and 2180.5 s runtime. These results show that the gain is not due to comparing only against a weak baseline: stronger optimizers do improve over single NSGA-II, **but the full AutoMS framework still performs best on this coupled cross-physics design problem**.
>
> The complete table is provided in the **[rebuttal for Reviewer tBx2, Q1 ](https://openreview.net/forum?id=zd6V4IqBkd&noteId=lBz7iXu9oW).**
>
> **Q3. Search Space and Physical Laws (Weakness 3 & Key Question 3)**
> Thank you for pointing this out.
> - Search Space: The search space is defined by the continuous latent mechanical conditioning space of the pre-trained MIND framework. Rather than randomly flipping geometric voxels, the agents optimize the input mechanical conditions (e.g., Target Young's Modulus, Poisson's ratio), which MIND then translates into a topological structure.
>
> - Ensuring Physical Validity: We do not rely on the LLM or the generator to strictly enforce physical laws, as they are prone to "hallucinations" in complex physics. Instead, physical validity is enforced a posteriori by the cross-physics simulator (FEA). If a generated design deviates from physical realism or target properties, the SAES loop utilizes the simulation feedback to correct the optimization trajectory and filter out invalid geometries. We will clarify this mechanism in Section 3.1 and 3.3.
>
> **Q4. Role of the LLM: Orchestration vs. Optimization (Key Question 5)**
>
> Thank you for this insightful question. The main challenge of inverse microstructure design is not only semantic understanding. It also involves a high-dimensional, discontinuous search space and expensive cross-physics simulations, where the optimization landscape is black-box and non-differentiable.
>
> Our design choice is therefore a deliberate division of labor, rather than an assumption that the task is mainly linguistic. In AutoMS, LLM-based agents are used where open-ended reasoning is most useful: decomposing user requirements, organizing multiple constraints, coordinating tools and agents, and deciding how the workflow should proceed. The Manager, Parser, Generator, Simulator, and Reporter form a closed-loop orchestration process.
>
> By contrast, the low-level optimization problem is fundamentally numerical: it must update conditioning variables under simulation feedback in a non-differentiable cross-physics environment. For this reason, we do not let agents directly control gradient estimation or parameter mutation. AutoMS intentionally separates two different kinds of difficulty:
> - open-ended requirement decomposition and workflow coordination, where agents are effective;
> - simulation-grounded numerical search, where dedicated optimization is more reliable.
>
> We will revise the manuscript to clarify this design rationale more explicitly.
>
> **Q5. Grammatical and Formatting Issues (Key Question 1)**
>
> Thank you for pointing this out. We have carefully proofread the manuscript and corrected the issue.

---

> > ### Author Rebuttal · Reviewer_5yWc · 2026-04-06
> >
> > The authors have tried their best to answer the questions I raised.
> > This work is meaningful and has some benefits for the field of computer science.
> > I still hope the authors can add more new comparison algorithms to optimize this paper, and I hope they can open-source their code in the final version.
> > I improved my score to 4.

---

> > > ### Author Response · Authors · 2026-04-06
> > >
> > > Dear Reviewer 5yWc,
> > >
> > > Thank you for your encouraging acknowledgement and for updating your score.
> > >
> > > We are glad that our rebuttal addressed your main concerns. In the revision, we will incorporate the newly added baselines, further strengthen comparisons with broader optimization methods, and release our code to support reproducibility.
> > >
> > > Thank you again for your constructive feedback.

---

### Official Review · Reviewer_CK1X · 2026-03-12

**Soundness:** 3
**Presentation:** 3
**Significance:** 3
**Originality:** 3
**Overall Recommendation:** 5
**Confidence:** 4

**Summary:**

The authors develop AutoMS, a multi-agent workflow that couples large-language models, finite element simulations, evolutionary search to design microstructures with target properties. The inverse design of microstructures is a challenging problem in material science, and the workflow proposed is quite interesting, and shows impressive success across many tasks. Specifically, AutoMS uses an orchestrator layer with an optimization layer where the optimization uses simulations and prior data to suggest candidates to the MIND framework, which generates microstructures.

**Compliance With Llm Reviewing Policy:**

Affirmed.

**Final Justification:**

No further comments from me - I would like to stick to prior assessment and my prior score.

**Key Questions For Authors:**

(1) I would encourage the authors to compare to Bayesian Optimization.
(2) I would encourage the authors to document the generative capabilities of MIND directly for one simple and one hard task, as a baseline for MIND’s capabilities and tendency of “hallucinate”.

**Limitations:**

yes

**Strengths And Weaknesses:**

Strengths
(1) The idea of integrating LLMs with local optimization that uses simulations as physical grounding is excellent for material science problems where surrogate models are not as readily available.
(2) Comparing to NSGA-II (a purely numerical optimization scheme) is a good baseline. The necessity of SAES is also clearly shown in Fig 4 and Table 3.
Weaknesses
(1) The state-of-the-art in materials design tasks seems to be Bayesian Optimization. How does AutoMS compare to a standard BayesOpt workflow?
(2) The authors use MIND to generate microstructures. While the need for constraining generators is clear, and the nature of generators to “hallucinate” is also believable, some benchmarking on this aspect would be useful. I am curious if some tasks can be solved by having MIND simply generate structures where the simulator “smooths over” the hallucinated features, converting them into realistic microstructures.
(3) Too many phrases in the text are in “quotes”.

---

> ### Author Rebuttal · Authors · 2026-03-31
>
> Thank you for your thoughtful review and insightful questions. Your suggestions for additional baselines and documentation of the generator's raw capabilities are highly valuable, and we have implemented them fully to strengthen the final manuscript.
>
> **Q1. Comparison to Bayesian Optimization and Other Solvers (Weakness 1 & Key Question 1)**
>
> We agree that Bayesian Optimization (BayesOpt) represents a standard and powerful workflow in materials design. To provide a more comprehensive evaluation, we have implemented a standard BayesOpt baseline. Additionally, to further solidify our benchmarking, we also integrated CMA-ES (Covariance Matrix Adaptation Evolution Strategy, a state-of-the-art gradient-free optimizer). We evaluated these baselines under the same simulation environment. **The result is shown in the rebuttal for Reviewer [tBx2, Q1](https://openreview.net/forum?id=zd6V4IqBkd&noteId=lBz7iXu9oW)**.
>
> The new results show that AutoMS remains superior to both numerical baselines on our 17-task suite. Specifically, **Bayesian Optimization achieves 65.0% SR, 0.0194 MRE, and 3982.9 s**, and **CMA-ES achieves 66.7% SR, 0.0241 MRE, and 4202.3 s, compared with 83.8% SR, 0.0140 MRE, and 2180.5 s for AutoMS**. These results suggest that while BayesOpt and CMA-ES are strong optimizers, they are less effective than AutoMS in the discontinuous, coupled cross-physics landscape considered here, where iterative simulation-grounded correction is especially important. We will add these baselines and the full comparison table to the revised manuscript.
>
>
> **Q2. Benchmarking MIND's Generative Capabilities & "Hallucinations" (Weakness 2 & Key Question 2)**
>
> Thank you for this helpful suggestion. To directly test whether MIND alone can solve these tasks, and whether the simulator could somehow “smooth over” unrealistic generations, we constructed a one-shot MIND baseline. This baseline uses the same MIND checkpoint, same target specification, and same simulator as AutoMS; the difference is that we remove SAES, iterative feedback, and correction. In other words, MIND receives the target once, generates one structure in a single pass, and that output is evaluated directly by the simulator.
>
> Going beyond the requested one simple and one hard case, we evaluated this baseline on the full 17-task suite. The result is that one-shot MIND succeeds on only 6/17 tasks and fails on the remaining 11/17, especially on tightly constrained cross-physics problems. **This shows that MIND does have useful raw generative ability, but that ability alone is not sufficient for reliable inverse design.**
>
> | Task | Difficulty | SR | CSR | MRE | BPM | QS |
> | :--- | :--- | :--- | :--- | :--- | :--- | :--- |
> | T1 | Hard | 0% | 0% | 0.6099 | 33% | 17.2 |
> | T2 | Hard | 0% | 0% | 0.6067 | 25% | 13.9 |
> | T3 | Easy | 100% | 100% | 0.0010 | 100% | 100.0 |
> | T4 | Medium| 0% | 0% | 0.0375 | 33% | 23.0 |
> | T5 | Medium | 0% | 80% | 0.0452 | 75% | 63.5 |
> | T6 | Medium | 100% | 100% | 0.0048 | 100% | 100.0 |
> | T7 | Medium | 100% | 100% | 0.0035 | 100% | 100.0 |
> | T8 | Medium | 100% | 80% | 0.0010 | 100% | 94.0 |
> | T9 | Hard | 0% | 75% | 0.3289 | 67% | 55.9 |
> | T10 | Medium | 0% | 0% | 0.0920 | 33% | 22.4 |
> | T11 | Hard | 0% | 0% | 1.0000 | 0% | 0.0 |
> | T12 | Medium | 100% | 100% | 0.0048 | 100% | 100.0 |
> | T13 | Medium | 100% | 100% | 0.0035 | 100% | 100.0 |
> | T14 | Hard | 0% | 0% | 0.4406 | 25% | 15.6 |
> | T15 | Hard | 0% | 0% | 0.0974 | 33% | 22.4 |
> | T16 | Easy | 0% | 0% | 0.8542 | 0% | 1.5 |
> | T17 | Easy | 0% | 25% | 0.0985 | 50% | 36.5 |
>
> To directly match your suggestion, we highlight one easy and one hard example. On an easy task such as T3, one-shot MIND reaches 100% SR with 0.0010 MRE. In contrast, on a tightly constrained hard task such as T2, it reaches 0% SR with 0.6067 MRE. **This contrast indicates that raw generation can occasionally succeed on simpler targets, but fails to do so robustly once the constraint coupling becomes stronger.**
>
> These results also clarify the role of the simulator. **The simulator is purely evaluative and does not modify or repair the generated geometry.** Therefore, it does not convert hallucinated one-shot generations into valid structures; instead, it exposes when a generated candidate is topologically or physically invalid, which must then be corrected by the iterative search loop. We will include the full 17-task one-shot MIND table in the revision to make this distinction explicit.
>
> **Q3. Stylistic Formatting (Weakness 3)**
>
> Thank you for pointing out the stylistic issue regarding the overuse of quotation marks. We have thoroughly reviewed the manuscript and removed unnecessary quotes around standard terms and phrases to improve the professional tone and readability of the text.

---

> > ### Author Rebuttal · Reviewer_CK1X · 2026-04-01
> >
> > Authors have answered my questions - I have deeper questions to ask and discuss, but I think that warrants a deep dive, and is beyond the scope of this paper. No change to my score since I've already accepted the paper.

---

> > > ### Author Response · Authors · 2026-04-03
> > >
> > > Dear Reviewer CK1X,
> > >
> > > Thank you very much for your encouraging acknowledgement. We sincerely appreciate your thoughtful feedback, which has helped us further strengthen the manuscript.
> > >
> > > We are excited to see how generative models and LLM-driven workflows are pushing the boundaries of structure generation, and we look forward to digging deeper in this direction.

---

### Official Review · Reviewer_RLGe · 2026-03-12

**Soundness:** 3
**Presentation:** 3
**Significance:** 3
**Originality:** 3
**Overall Recommendation:** 4
**Confidence:** 3

**Summary:**

This paper introduces AutoMS an optimization/design framework for designing microstructures under cross physics objectives. The problem represents an inverse design challenge due to large design spaces and non-differentiability of the objective measure. The method reformulates the problem as evolutionary search guided by a set of LLM agents that receive feedback from physical simulation. The main innovation appears to be Simulation-Aware Evolutionary Search (SAES), which employs local gradient approximation to guide LLM agents in iterating on the design population. The author's evaluated their method on a number of tasks and found it superior to a multi-objective evolutionary search (NSGA-II) baseline and a single agent baseline.

**Compliance With Llm Reviewing Policy:**

Affirmed.

**Key Questions For Authors:**

- Can you provide one or more stronger evolutionary baselines?
- If MIND was replaced with a weaker generator how much better is AutoMS than the baselines?
- The full system shows lower CSR than the version that ablates SAES. Can you go deeper into this? In what task domains or levels of difficulty does the lower CSR actually become a liability?
- Can you provide statistical confidence measures for the reported performance scores?
- How much compute time is spent on LLM inference versus the simulation engine?

**Limitations:**

The authors do address a number of limitations, mostly related to sim2real transferability of the discovered designs. However, the paper should more clearly discuss:
- computational expense of using LLMs for inference in the context of this system (i.e. the system's scalability)

**Strengths And Weaknesses:**

## Strengths
- The paper is well written, highly accessible.
- The problem is well motivated: the difficult search landscape likely necessitates the development of new algorithms. The idea of iterative refinement in this domain is compelling when considering the hallucinated failures of one-shot approaches.
- The use of an indirect representation for evolution likely increases the capabilities of the system.
- The evaluation suite appears thorough: it includes a variety of different tasks and difficulty levels, and the authors performed a number of ablation studies illustrating the impact of their method's distinct components.

## Weaknesses
There exist several weaknesses that need to be addressed:
- The selected baseline NSGA-II appears to be very weak. CMA-ES is known to be a superior black-box gradient-free optimizer. Without comparison to modern, SOTA evolutionary methods, the authors cannot claim to be superior to evolutionary methods. This is my primary concern and why I provided a Soundness rating of 2.
- The task suite was designed entirely by the authors. Are there no community benchmarks that could be used in this setting? That would strengthen the method as it would become much easier to compare to prior work.
- Confidence reporting: the paper does not appear to report standard deviations or confidence intervals for key performance metrics. This makes it difficult to understand the empirical usability of the system and also contributed to my Soundness rating.
- The system is built on top of MIND, but there is no ablation of this core generator. If the authors use another (perhaps known to be weaker) generator they could decouple the performance of AutoMS.

---

> ### Author Rebuttal · Authors · 2026-03-31
>
> Thank you for your thoughtful review and insightful questions. We have addressed each point below and will incorporate the corresponding clarifications and additional results in the revision.
>
> **Q1. Stronger Evolutionary Baselines (Weakness 1 & Key Question 1)**
>
> You raised an excellent point regarding the selection of NSGA-II. We agree that CMA-ES is a superior optimizer. To rigorously prove the superiority of AutoMS over modern evolutionary methods, we further evaluated a strong CMA-ES baseline on the full benchmark suite. As shown in the new results below, while CMA-ES outperforms NSGA-II, AutoMS with SAES still achieves better overall performance in the challenging cross-physics search landscape, reaching 83.8% SR and 0.0140 MRE, compared with 66.7% SR and 0.0241 MRE for CMA-ES.
>
> The complete table is provided in the **[rebuttal for Reviewer tBx2, Q1 ](https://openreview.net/forum?id=zd6V4IqBkd&noteId=lBz7iXu9oW).**
>
> **Q2. Generator Ablation (Weakness 4 & Key Question 2)**
>
> We agree that decoupling AutoMS from the underlying generator is important for a rigorous evaluation. We therefore replaced MIND with a weaker generator ([Yang et al., 2026], [1]) while keeping the rest of the AutoMS framework unchanged. Under this weaker generator, performance drops from 83.8% to 73.3% SR and from 94.2% to 88.9% BPM, while runtime increases from 2180.5 s to 4223.3 s, confirming that the generator quality substantially affects both design quality and efficiency.
> At the same time, the results also show that AutoMS is not tied to MIND as a specific backbone: even with a weaker generator, AutoMS remains competitive and still outperforms several stronger baselines, including CMA-ES (66.7% SR), Bayesian Optimization (65.0% SR), and Single Agent + SAES (69.2% SR), while remaining close to CoT+SAES (74.1% SR). This suggests that the performance gains of AutoMS do not come solely from the underlying generator, but also from the effectiveness of the proposed closed-loop multi-agent optimization framework.
>
> | Method | SR | CSR | MRE | BPM | QS | Iter | Time |
> | :--- | :--- | :--- | :--- | :--- | :--- | :--- | :--- |
> | MIND | 83.8% | 59.6% | 0.0140 | 94.2% | 82.4 | 14.4 | 2180.5s |
> | [Yang et al., 2026] | 73.3% | 57.7% | 0.0137 | 88.9% | 80.7 | 12.3 | 4223.3s |
>
>
> **Q3. The CSR vs. SR Trade-off (Key Question 3)**
>
> The lower CSR is expected and reflects the role of SAES. **CSR counts partially satisfied candidates, while SR counts fully valid solutions**. Without SAES, the optimizer more easily converges to easier partial matches, especially on harder tasks with tightly coupled objectives and narrow feasible regions; this can inflate CSR while lowering SR. With SAES, AutoMS is guided away from these partial local optima toward fully feasible cross-physics designs. We will revise Section 5.3 to clarify this distinction and mechanism.
>
> **Q4. Community Benchmarks (Weakness 2)**
>
> We agree that standardized benchmarks make comparisons much easier. However, the inverse design of 3D microstructures under coupled cross-physics constraints is a highly emerging domain. Currently, there are no established community benchmarks that provide multi-objective, cross-physics targets paired with high-fidelity validation environments. Because of this gap, we were compelled to design a comprehensive 17-task suite covering various materials, modalities, and difficulty stratifications. We hope our task suite can serve as a foundational benchmark for future researchers in this domain.
>
> **Q5. Statistical Confidence Measures (Weakness 3 & Key Question 4)**
>
> We thank the reviewer for pointing this out. Our reported results are aggregated over the full 17-task benchmark suite, and each task was evaluated with four independent runs to ensure statistical robustness. As shown in the table below, our core metrics exhibit low variance, indicating high algorithmic stability.
>
> |Metric|Mean |Standard Deviations|
> |:------ |------:|-------:|
> |SR|83.8%|3.3%|
> |CSR| 59.6%|13.1%|
> |MRE|0.0140|0.0072|
> |BPM|94.2%|1.7%|
> |QS|82.4|4.67|
> |Iter|14.4|8.05|
> |Time(s)|2180.5|1346.13|
>
> **Q6. Compute Time Breakdown & Scalability (Key Question 5 & Limitations)**
>
> To address your concerns regarding the computational expense of LLM inference versus the simulation engine, we have profiled the system's resource consumption.
> The Orchestration Layer acts as a computational gatekeeper, preventing wasteful simulation cycles. We will include the following detailed breakdown in the revision to explicitly discuss scalability:
> |Metric|Expense|
> |:---|:---|
> |**Total Wall-clock Time (avg per task)**|**2180.5 s**|
> |LLM Inference Time | 669.4 s (30.7%)|
> |Simulation Engine / Tool Execution Time|1508.9 s (69.2%)|
> | **Average Token Consumption per Task**|**457306 tokens**|
> |Input tokens|436911|
> |Output tokens|20395|
>
> [1] Guided diffusion for fast inverse design of voxel-based mechanical metamaterials. Smart Materials in Manufacturing, 2026.

---

> > ### Author Rebuttal · Reviewer_RLGe · 2026-03-31
> >
> > I thank the authors for the thorough rebuttal. The addition of stronger baselines (CMA-ES, BayesOpt), the generator ablation, statistical confidence measures, and compute breakdown have adequately addressed my concerns. I am raising my Soundness score to 3 and my overall recommendation to 4.

---

> > > ### Author Response · Authors · 2026-04-03
> > >
> > > Dear Reviewer RLGe,
> > >
> > > We sincerely thank you for your encouraging acknowledgement!
> > >
> > > We are delighted that our rebuttal has adequately addressed your concerns. We will incorporate these additional results and clarifications in the revised manuscript.
> > >
> > > Thank you again for your highly constructive feedback and helping us improve our work.

---

### Official Review · Reviewer_tBx2 · 2026-03-13

**Soundness:** 3
**Presentation:** 1
**Significance:** 3
**Originality:** 3
**Overall Recommendation:** 4
**Confidence:** 4

**Summary:**

This paper introduces a multi-agent approach for cross-physics inverse microstructure design. The approach is well designed, consisting of two layers that focus on orchestration and optimization, respectively, with clear motivations. In the optimization layer, a simulation-aware evolutionary search is introduced with three sequential steps: local gradient approximation, parameter update, and Pareto-driven selection. The experimental results show that AutoMS performs well compared with NSGA-II and a ReAct-based LLM.

**Compliance With Llm Reviewing Policy:**

Affirmed.

**Final Justification:**

From my perspective, the approach is not sufficiently novel, as it primarily combines an agentic design with NSGA-II. However, the paper is interesting and the problem it addresses is promising. During the rebuttal, the authors had addressed many of my concerns. Therefore, I would like to increase my score.

**Key Questions For Authors:**

See weaknesses and limitations.

**Limitations:**

The experiments are generally well-conducted. However, several important experiments are missing to clearly demonstrate the contribution of each component of AutoMS and to strengthen the comparisons with other methods.

First, the comparison experiments are relatively limited, as the paper only compares AutoMS with two methods. A broader set of baselines would provide a more convincing evaluation. For example, it would be useful to consider optimization methods such as genetic algorithms, CMA-ES, and NSGA-II, combined with multi-agent frameworks. In addition, SAES integrated with CoT prompting, LLM reasoning, or agent-based designs could also serve as meaningful baselines.

Second, in the ablation study, it would be preferable to continue using NSGA-II rather than replacing the optimization layer with a genetic algorithm. Maintaining the same optimization framework would allow the effect of each component in AutoMS to be evaluated more clearly and would make the ablation results easier to interpret.

One major concern relates to mutation and crossover. These operators are mentioned in Line 96, but the paper does not appear to provide further details in the method section. As mutation and crossover are core operations in evolutionary algorithms, the paper should clearly describe how they are implemented within the proposed framework. Without this information, it is difficult to fully understand or reproduce the proposed method.

**Strengths And Weaknesses:**

Strengths:
1.	Overall, the approach is well-designed, and each component is clearly motivated.

2.	The perception–action–integration process, corresponding to local gradient approximation, parameter update, and Pareto-driven selection, is interesting and represents a heuristic-informed design.

Weaknesses:
1.	The paper title mentions multi-agent. However, from my perspective, the discussions of the different agents are very limited. This makes the approach appear to rely primarily on SAES, which seems to make the primary contribution rather than the agent framework.

2.	Regarding Table 1, it seems unusual that NSGA-II has Iter = 48.8. The iteration count is defined as the number of agent interactions, yet Line 315 states that NSGA-II operates without agent collaboration. This raises a potential inconsistency in how the iteration metric is defined. In addition, it is unclear why the iteration value contains decimals, since iteration counts are typically integers.

3.	Lines 19 and 26, in latex, the quotation mark should be ``, instead of ”. This is applied for the rest of the paper.

4.	In Line 133, it would be better to clarify the Ω, I guess it is microstructure geometry space.

5.	∇f is not defined clearly when it shows up.

6.	In Line 221, $x_k$ means the design, in fact it denotes the variables as defined in section 3.1. It would be better to clarify this.
7.	In Line 226, y(x) seems not clear enough; y is the simulated property, but it represents the function at the same time.

---

> ### Author Rebuttal · Authors · 2026-03-31
>
> Thank you for your thoughtful review and insightful questions. We have carefully addressed concerns below and will incorporate all clarifications into the revised manuscript.
>
> **Q1. Expanded Baselines and Evaluation (Limitations 1)**
>
> We completely agree that a broader set of baselines strengthens the evaluation. Following your suggestion, we have conducted new experiments replacing our core modules with **CMA-ES, NSGA-II, and CoT reasoning** to isolate the performance of each component.
> We will include the following updated results in the revised manuscript:
>
> **More  baseline comparisons**
>
> | Method | SR | CSR | MRE | BPM | QS | Iter | Time |
> |:---|:---|:---|:---|:---|:---|:---|:---|
> |CMA-ES|66.7%|55.7%|0.0241|84.7%|73.7|40.8|4202.3s|
> |Bayesian Optimization | 65.0% | 55.6% | 0.0194 | 85.6% | 73.73 | 39.2 | 3982.9s |
> |Single Agent + SAES | 69.2% | 66.4% | 0.0172 | 86.5% | 78.19 | 9.6 | 1743.0s |
> |CoT+SAES| 74.1% | 70.2% | 0.0227 | 90.5% | 81.9 | 16.7 | 2980.8s |
>
>
> **More  ablation studies**
>
> | Method | SR | CSR | MRE | BPM | QS | Iter | Time |
> | :--- | :--- | :--- | :--- | :--- | :--- | :--- | :--- |
> | AutoMS w/ reasoner | 75.0% | 66.7% | 0.0154 |89.7% | 80.74 | 18.1 | 3653.8s |
> | AutoMS w/o NSGA-II w/ GA | 76.9% | 61.0% | 0.0227 | 84.7% | 77.33 | 15.12 | 2767.4s |
> | AutoMS w/o NSGA-II w/ CMA-ES | 81.7% | 69.6% | 0.0158 | 86.9% | 81.82 | 15.2 | 2696.6s |
> | AutoMS | 83.8% | 59.6% | 0.0140 | 94.2% | 82.4 | 14.4 | 2180.5s |
>
>
> These added baselines let us isolate the contribution of each component more cleanly:
> - AutoMS vs. [CMA-ES] / [Bayesian Optimization] tests against strong non-LLM optimization baselines;
> - AutoMS vs. [Single-Agent + SAES] / [CoT + SAES] tests the value of multi-agent decomposition;
> - AutoMS vs. [AutoMS w/o NSGA-II w/ GA]  / [AutoMS w/o NSGA-II w/ CMA-ES] tests the value of SAES as the optimization module.
>
> We will include these results in the revision and update the discussion accordingly.
>
>
> **Q2. Clarification on the Multi-Agent Contribution (Weakness 1)**
>
> We appreciate this question, because it goes to the core of the paper. The contribution of AutoMS is not that SAES alone improves search; rather, **the main contribution is the closed-loop multi-agent decomposition, with SAES serving as one key optimization module inside that loop.** The paper’s architecture explicitly separates an Orchestration Layer (Manager, Parser, Generator, Simulator, Reporter) from the Optimization Layer, where SAES provides simulation-aware search updates.
>
> More importantly, the new controlled results in Q1 directly show that the gains are not reducible to SAES alone. Under the same search module family, AutoMS outperforms Single-Agent + SAES (83.8 vs. 69.2 SR) and CoT + SAES (83.8 vs. 74.1 SR). This isolates the effect of role-specialized coordination: the Parser resolves semantic ambiguity into executable targets, the Generator proposes candidates, the Simulator grounds the loop with cross-physics validation, and the Manager/Reporter coordinate and summarize decisions across iterations. **SAES improves the numerical search, but the multi-agent structure is what makes the end-to-end semantic-to-physical loop reliable.** We will revise Section 3.2 to make this separation and the supporting evidence explicit.
>
>
> **Q3. Clarification on Iteration Metric and NSGA-II (Weakness 2)**
>
> Thank you for pointing this out. The 48.8 reported for NSGA-II is **the average number of evolutionary generations to converge across four runs**. More generally, the Iter metric is a method-specific convergence-cycle proxy: for AutoMS/ReAct-style systems it counts interaction/search rounds, whereas for evolutionary baselines it counts generations.
>
>
> **Q4. Correction Regarding the Ablation Study (Limitations 2)**
>
> We clarify that this was a writing error. In the “w/o SAES” ablation of Table 3, we kept NSGA-II as the optimization framework for consistency, rather than swapping to a standard genetic algorithm. To avoid ambiguity, the new GA/CMA-ES rows in Q1 replace the optimizer entirely, while the original w/o SAES ablation keeps NSGA-II selection and removes only the SAES update.
>
> **Q5. Details on Mutation and Crossover Implementation (Limitations 3)**
>
> AutoMS does not optimize discrete voxels directly; it optimizes continuous mechanical conditioning variables. Accordingly, both mutation and crossover are implemented in this conditioning space.
> Mutation follows the simulation-aware update in Eq. (5), combining a gradient-guided step with adaptive Gaussian noise. Crossover samples parent conditioning vectors from the elite archive and applies simulated binary crossover. We will clarify this in the appendix.
>
> **Q6. Addressing Typos and Formatting (Weaknesses 3-7)**
>
> Thank you for the careful review.  We have corrected the quotation marks throughout the manuscript. Furthermore, we have explicitly clarified the definitions of $\Omega$, $\nabla f$, and $y(x)$ to eliminate any notational ambiguity.

---

> > ### Author Rebuttal · Reviewer_tBx2 · 2026-04-02
> >
> > Thanks for the authors’ rebuttal. The additional experiments are helpful in addressing my previous concerns. However, I still have reservations regarding the effectiveness and efficiency of the proposed method compared with existing approaches, especially considering the performance of CMA-ES. Therefore, I would like to maintain my original score.

---

> > > ### Author Response · Authors · 2026-04-03
> > >
> > > Dear Reviewer tBx2,
> > >
> > > Thank you for your continued engagement and for sharing your remaining reservations. We realize that our previous table may have caused confusion by presenting the standalone CMA-ES baseline and the internal CMA-ES ablation together. We would like to clarify this critical distinction, which directly addresses your concerns regarding **effectiveness and efficiency**:
> > >
> > > 1. **Standalone CMA-ES (The Existing Approach Baseline)**
> > >  This is the standard, end-to-end CMA-ES without our multi-agent framework or simulation-aware gradient guidance.
> > > - Performance: SR = 66.7%, Time = 4202.3s
> > >
> > > 2. **AutoMS w/o NSGA-II w/ CMA-ES (Our Internal Ablation)**
> > >  This is our proposed AutoMS framework (retaining the multi-agent loop and the SAES gradient-guided updates), but replacing only the NSGA-II selection component with CMA-ES.
> > > - Performance: SR = 81.7%, Time = 2696.6s
> > >
> > > 3. **Full AutoMS (Our Proposed Method)**
> > > - Performance: SR = 83.8%, Time = 2180.5s
> > >
> > > **Effectiveness & Efficiency Takeaway**:
> > > When comparing AutoMS to the actual existing approach (Standalone CMA-ES), AutoMS demonstrates a **substantial 17.1% absolute improvement in Success Rate (83.8% vs. 66.7%)** and is **nearly 2x faster (2180.5s vs. 4202.3s)**. This clearly establishes the superior effectiveness and efficiency of our method.
> > >
> > > The fact that our internal ablation (Variant 2) performs well does not imply that CMA-ES is equivalent to AutoMS. Instead, it strongly supports our core claim: the significant performance gains come primarily from our proposed **multi-agent closed-loop design** and the **simulation-aware gradient updates**. Because these core innovations are retained in Variant 2, the performance remains high, proving that our framework is robust and not narrowly dependent on NSGA-II alone.
> > >
> > > We deeply appreciate your feedback, which helped us realize that this distinction was not sufficiently explicit. In the revision, we will clearly separate external optimizer baselines from internal optimizer-replacement ablations to make the effectiveness/efficiency tradeoff unambiguous. We hope this clarification fully resolves your remaining reservations.

---

### Decision · Program_Chairs · 2026-04-30

**Decision:**

Accept (regular)

**Comment:**

The paper studies an interesting inverse design problem with a well-structured multi-agent framework. Reviewer tBx2 became more positive after the rebuttal clarified the multi-agent role and added stronger baselines, while Reviewer RLGe and Reviewer CK1X both found the main concerns addressed by the new CMA-ES/BayesOpt results and the added generator analysis. Although some scope and positioning issues remain, I think the paper is enough for acceptance.